# GPIHBP1 expression in gliomas promotes utilization of lipoprotein-derived nutrients

Xuchen Hu[1], Ken Matsumoto[2], Rachel S Jung[1], Thomas A Weston[1], Patrick J Heizer[1], Cuiwen He[1], Norma P Sandoval[1], Christopher M Allan[1], Yiping Tu[1], Harry V Vinters[3], Linda M Liau[4,5], Rochelle M Ellison[1], Jazmin E Morales[1], Lynn J Baufeld[6,7], Nicholas A Bayley[6,7], Liqun He[8], Christer Betsholtz[8,9], Anne P Beigneux[1], David A Nathanson[6,7], Holger Gerhardt[2,10], Stephen G Young[1,11]*, Loren G Fong[1]*, Haibo Jiang[1,12]*

[1]Department of Medicine, David Geffen School of Medicine, University of California, Los Angeles, Los Angeles, United States; [2]VIB-KU Leuven Center for Cancer Biology (CCB), Leuven, Belgium; [3]Department of Pathology and Laboratory Medicine, David Geffen School of Medicine, University of California, Los Angeles, Los Angeles, United States; [4]Department of Neurosurgery, David Geffen School of Medicine, University of California, Los Angeles, Los Angeles, United States; [5]Jonsson Comprehensive Cancer Center (JCCC), David Geffen School of Medicine, University of California, Los Angeles, Los Angeles, United States; [6]Department of Molecular and Medical Pharmacology, David Geffen School of Medicine, University of California, Los Angeles, Los Angeles, United States; [7]Ahmanson Translational Imaging Division, David Geffen School of Medicine, University of California, Los Angeles, Los Angeles, United States; [8]Department of Immunology, Genetics and Pathology, Rudbeck Laboratory, Uppsala University, Uppsala, Sweden; [9]Integrated Cardio Metabolic Centre (ICMC), Karolinska Institutet, Huddinge, Sweden; [10]Max Delbrück Center for Molecular Medicine, Berlin, Germany; [11]Department of Human Genetics, David Geffen School of Medicine, University of California, Los Angeles, Los Angeles, United States; [12]School of Molecular Sciences, University of Western Australia, Perth, Australia

*For correspondence:
sgyoung@mednet.ucla.edu (SGY);
lfong@mednet.ucla.edu (LGF);
haibo.jiang@uwa.edu.au (HJ)

**Abstract** GPIHBP1, a GPI-anchored protein of capillary endothelial cells, binds lipoprotein lipase (LPL) within the subendothelial spaces and shuttles it to the capillary lumen. GPIHBP1-bound LPL is essential for the margination of triglyceride-rich lipoproteins (TRLs) along capillaries, allowing the lipolytic processing of TRLs to proceed. In peripheral tissues, the intravascular processing of TRLs by the GPIHBP1–LPL complex is crucial for the generation of lipid nutrients for adjacent parenchymal cells. GPIHBP1 is absent from the capillaries of the brain, which uses glucose for fuel; however, GPIHBP1 is expressed in the capillaries of mouse and human gliomas. Importantly, the GPIHBP1 in glioma capillaries captures locally produced LPL. We use NanoSIMS imaging to show that TRLs marginate along glioma capillaries and that there is uptake of TRL-derived lipid nutrients by surrounding glioma cells. Thus, GPIHBP1 expression in gliomas facilitates TRL processing and provides a source of lipid nutrients for glioma cells.
DOI: https://doi.org/10.7554/eLife.47178.001

## Introduction

GPIHBP1, a GPI-anchored protein of capillary endothelial cells, is required for lipoprotein lipase (LPL)–mediated processing of triglyceride-rich lipoproteins (TRLs) (*Beigneux et al., 2007*).

The principal function of GPIHBP1 is to capture LPL within the interstitial spaces, where it is secreted by parenchymal cells, and then to shuttle this enzyme to the luminal surface of capillary endothelial cells (*Davies et al., 2010*). GPIHBP1 is a long-lived protein (*Young et al., 2011*; *Olafsen et al., 2010*) that moves bidirectionally across endothelial cells, with each trip to the abluminal plasma membrane representing an opportunity to capture LPL and bring it to the capillary lumen (*Davies et al., 2012*). When GPIHBP1 is absent or defective, LPL is stranded within the interstitial spaces, where it remains bound to sulfated proteoglycans near the surface of cells (*Young et al., 2011*; *Davies et al., 2010*; *Allan et al., 2017a*; *Fong et al., 2016*). The inability of LPL to reach the capillary lumen in the absence of GPIHBP1 expression profoundly impairs TRL processing, resulting in severe hypertriglyceridemia (chylomicronemia) (*Beigneux et al., 2007*; *Davies et al., 2010*; *Goulbourne et al., 2014*).

GPIHBP1 is expressed in the capillary endothelial cells of peripheral tissues, with particularly high levels of expression in heart and brown adipose tissue (*Beigneux et al., 2007*; *Davies et al., 2010*; *Fong et al., 2016*). Most of the LPL within those tissues is bound to GPIHBP1 on capillaries (*Beigneux et al., 2007*; *Davies et al., 2010*; *Davies et al., 2012*; *Allan et al., 2017a*; *Fong et al., 2016*; *Allan et al., 2017b*; *Allan et al., 2016*), and the processing of TRLs in these tissues is robust, generating fatty acid nutrients for nearby parenchymal cells (*Fong et al., 2016*; *Jiang et al., 2014a*; *He et al., 2018a*). By contrast, GPIHBP1 is absent from capillaries of the brain (*Young et al., 2011*; *Davies et al., 2010*; *Olafsen et al., 2010*), a tissue that depends on glucose for fuel (*Mergenthaler et al., 2013*). When wild-type mice are injected intravenously with a GPIHBP1-specific antibody, the antibody rapidly binds to GPIHBP1-expressing capillaries in peripheral tissues and disappears from the plasma (*Davies et al., 2010*; *Olafsen et al., 2010*). By contrast, there is no antibody binding to the capillaries of the brain (*Davies et al., 2010*; *Olafsen et al., 2010*).

For the lipolytic processing of TRLs to proceed, lipoproteins in the bloodstream must marginate along the luminal surface of capillaries (*Goulbourne et al., 2014*). TRL margination along capillaries depends on GPIHBP1, more specifically on GPIHBP1-bound LPL (*Goulbourne et al., 2014*). In GPIHBP1-deficient mice, TRLs never stop along heart capillaries and instead simply 'flow on by' in the bloodstream (*Goulbourne et al., 2014*). In wild-type mice, TRLs marginate along heart capillaries, but TRL margination is absent along capillaries of the brain (*Goulbourne et al., 2014*).

Even though GPIHBP1 is not found in brain capillaries, there is ample evidence for LPL expression within the brain (*Ben-Zeev et al., 1990*; *Bessesen et al., 1993*; *Goldberg et al., 1989*; *Vilaró et al., 1990*; *Yacoub et al., 1990*; *Eckel and Robbins, 1984*). Several groups have found LPL in the rat brain, specifically in neurons of the dentate gyrus and hippocampus, in pyramidal cells of the cortex, and in Purkinje cells of the cerebellum (*Ben-Zeev et al., 1990*; *Bessesen et al., 1993*; *Goldberg et al., 1989*; *Vilaró et al., 1990*; *Eckel and Robbins, 1984*). Using single-cell RNA sequencing, *Zhang et al. (2014)* found *Lpl* transcripts in the resident macrophages of the brain (microglia), with lower levels in astrocytes, neurons, and oligodendrocytes. Using the same approach, *Vanlandewijck et al. (2018)* found LPL expression in brain smooth muscle cells and in perivascular fibroblasts (at even higher levels than in microglial cells). Given the absence of GPIHBP1 expression in brain capillaries and the absence of TRL margination along brain capillaries, we have proposed that the LPL in the brain probably has an extravascular function, presumably to hydrolyze glycerolipids within the extracellular spaces (*Young et al., 2011*; *Adeyo et al., 2012*).

Despite the absence of GPIHBP1 expression in brain capillaries, we were curious about whether GPIHBP1 might be expressed in the capillaries of gliomas. Glioma capillaries are morphologically distinct from normal brain capillaries (*Yuan et al., 1994*; *Hobbs et al., 1998*; *Monsky et al., 1999*; *Bullitt et al., 2005*), and the blood–brain barrier is often defective (*Zhang et al., 1992*). Electron microscopy has suggested that glioblastoma capillaries resemble capillaries in peripheral tissues (*Vaz et al., 1996*).

If GPIHBP1 were to be expressed in glioma capillaries, it could be relevant to glioma metabolism. The GPIHBP1 might capture locally produced LPL, allowing for TRL margination and TRL processing, and thereby providing a source of lipid nutrients for glioma cells. Interestingly, *Dong et al. (2017)* documented LPL expression in gliomas. Also, several studies have raised the possibility that glioma cells use fatty acids for fuel (*Lin et al., 2017*; *Guo et al., 2011*; *Guo et al., 2009a*; *Guo et al., 2009b*; *Guo et al., 2013*) and that levels of free fatty acids are higher in gliomas than in normal brain tissue (*Guo et al., 2013*; *Gopal et al., 1963*).

In the current study, we sought to determine whether glioma capillaries express GPIHBP1 and, if so, whether it binds LPL and facilitates TRL margination and the lipolytic processing of TRLs. In our study, we took advantage of NanoSIMS imaging, a high-resolution mass spectrometry–based imaging modality that makes it possible to visualize TRL margination and TRL processing in tissue sections (*He et al., 2018a*; *Jiang et al., 2014a*; *Jiang et al., 2014b*; *He et al., 2017a*; *He et al., 2017b*; *He et al., 2018b*; *He et al., 2018c*). This imaging modality allowed us to visualize TRL margination in glioma capillaries as well as the entry of TRL-derived nutrients into tumor cells.

## Results

### GPIHBP1 is expressed in the endothelial cells of human gliomas

We sectioned 20 human gliomas (*Table 1*) and screened them for GPIHBP1 expression by confocal microscopy with three GPIHBP1-specific monoclonal antibodies (mAbs) — RF4, which binds to residues 27–44 downstream from GPIHBP1's acidic domain (*Kristensen et al., 2018*); and RE3 and RG3, which both bind to GPIHBP1's LU (Ly6/uPAR) domain (*Hu et al., 2017*). GPIHBP1 in capillary endothelial cells was detected in 14 of 20 gliomas (*Table 1*) and colocalized with von Willebrand factor, an endothelial cell marker (*Figure 1*). GPIHBP1 expression in glioma capillaries did not appear to correlate with glioma grade, 1p/19q co-deletions, or IDH1 mutations (*Table 1*). GPIHBP1 was not detectable in the capillaries of human brain specimens (*Figure 1*). The GPIHBP1 in glioma capillaries could be detected with all three GPIHBP1-specific mAbs (*Figure 2A*). To be confident in the

**Table 1.** Human glioma tumor specimens.
Expression of GPIHBP1 was assessed by immunohistochemistry with mAbs against human GPIHBP1 (RF4, RE3, RG3). Those conducting the studies were blinded to diagnoses. This table details the tumor diagnosis, location, 1p/19q co-deletion, and IDH1 mutation status, as well as the presence of GPIHBP1.

| Sample ID | Tissue diagnosis | Location | 1p/19q co-deletion | IDH1 mutation | GPIHBP1 |
|---|---|---|---|---|---|
| 1 | Glioblastoma (GBM) | Right frontal, parietal | No | Negative | Yes |
| 2 | GBM | Left temporal | No | Negative | Yes |
| 3 | GBM | Right occipital | No | Negative | Yes |
| 4 | GBM | Left frontal | No | Negative | Yes |
| 5 | Oligodendroglioma Grade II | Left anterior temporal, left posterior temporal | Yes | Negative | Yes |
| 6 | Oligoastrocytoma Grade III | Right temporal | No | Negative | Yes |
| 7 | GBM + oligodendroglial component | Left frontal | Yes | Negative | Yes |
| 8 | GBM + extensive oligodendroglial component | Right frontal | No | Negative | Yes |
| 9 | Oligodendroglioma Grade III | Left frontal | Yes | +R132H | Yes |
| 10 | Oligodendroglioma Grade III | Left frontal | Yes | +R132H | Yes |
| 11 | Oligoastrocytoma | Right parietal | No | Negative | Yes |
| 12 | Oligodendroglioma Grade III | Right parietal | Yes | +R132H | Yes |
| 13 | Oligodendroglioma Grade III | Right parietal | Yes | Negative | Yes |
| 14 | Oligoastrocytoma Grade III | Left temporal | No | +R132H | Yes |
| 15 | Oligoastrocytoma Grade III | Right temporal | No | +R132G | No |
| 16 | Oligoastrocytoma Grade III | Right frontal | No | +R132H | No |
| 17 | Oligodendroglioma Grade III | Left frontal | Yes | Negative | No |
| 18 | Oligodendroglioma Grade III | Left frontal | Yes | +R132H | No |
| 19 | Oligodendroglioma Grade III | Left temporal | Yes | Negative | No |
| 20 | Oligodendroglioma Grade III | Right temporal | Yes | +R132H | No |

DOI: https://doi.org/10.7554/eLife.47178.005

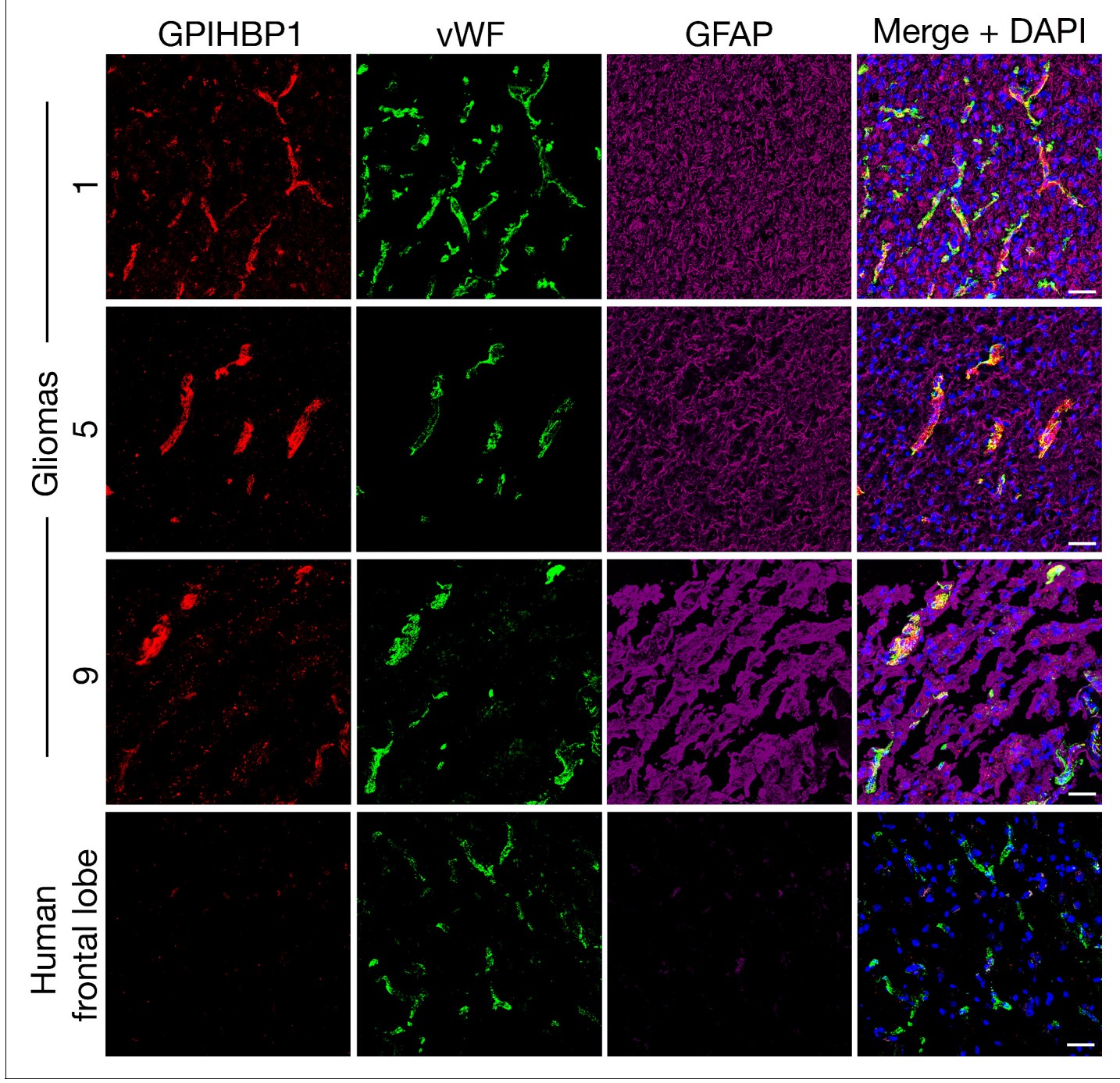

**Figure 1.** GPIHBP1 expression in the endothelial cells of several human gliomas. Immunohistochemical studies on surgically resected gliomas (Gliomas 1, 5, 9; *Table 1*) and non-diseased human frontal lobe (n = 3), revealing GPIHBP1 expression in capillaries of gliomas but not in frontal lobe specimens. GPIHBP1 (detected with a combination of the mAbs RE3 and RF4, 10 μg/ml each [red]) colocalized with von Willebrand factor (vWF, a marker for endothelial cells [green]), but not with glial fibrillary acidic protein (GFAP, a marker for astroglial cells [magenta]). DNA was stained with DAPI (blue). Three sections of each tumor and normal brain were evaluated and representative images are shown. Scale bar, 50 μm.

DOI: https://doi.org/10.7554/eLife.47178.002

The following figure supplement is available for figure 1:

**Figure supplement 1.** Detecting GPIHBP1 in glioma capillaries with immunoperoxidase staining.

DOI: https://doi.org/10.7554/eLife.47178.003

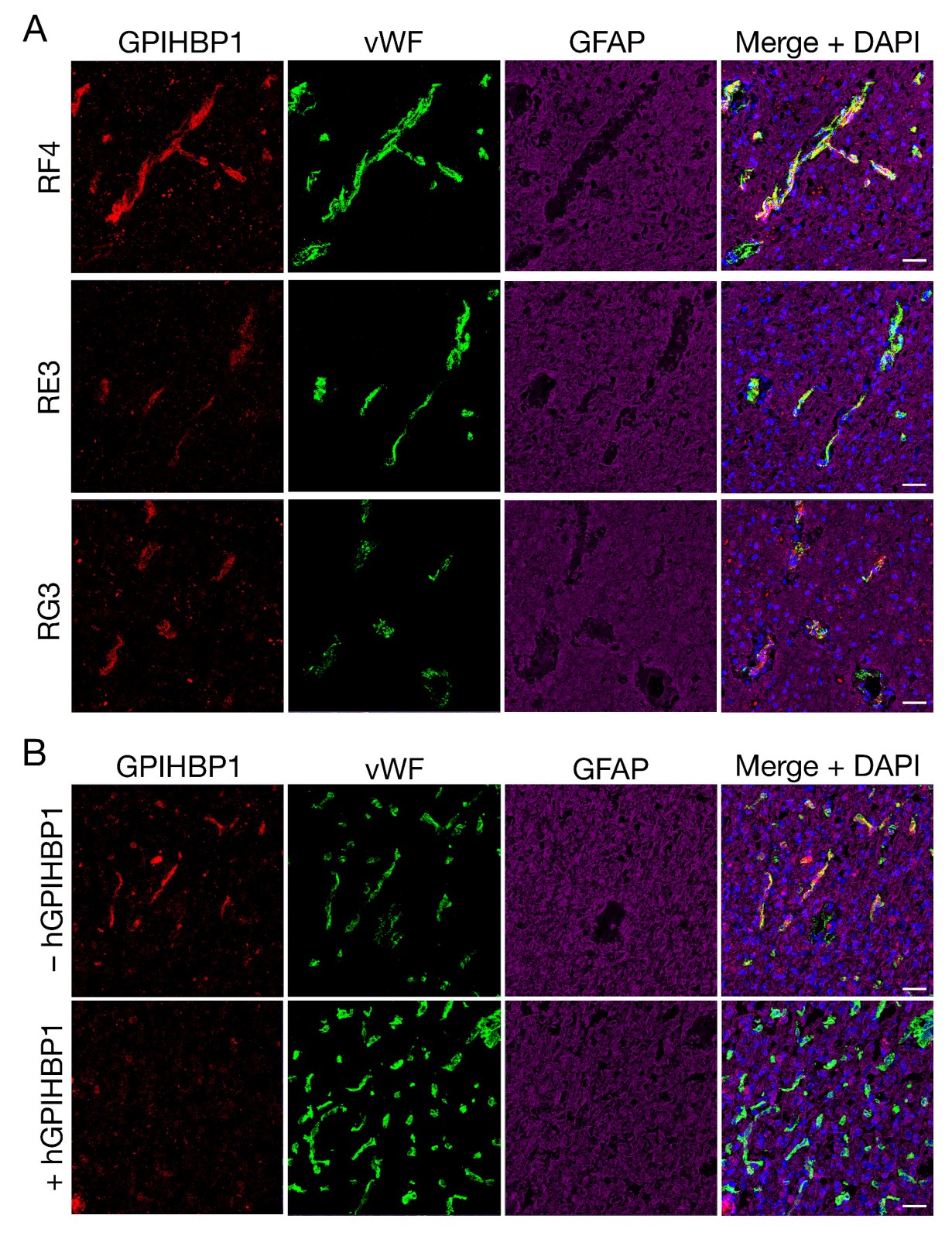

**Figure 2.** Detecting GPIHBP1 in capillaries of human glioma specimens with three different monoclonal antibodies (mAbs) against GPIHBP1. (**A**) Confocal fluorescence microscopy studies on sections from glioma sample 1 (**Table 1**), demonstrating the detection of GPIHBP1 with three different human GPIHBP1–specific monoclonal antibodies (mAbs). Tissue sections were fixed with 3% PFA and then stained with mAbs against human GPIHBP1 (RF4, RE3, or RG3, 10 µg/ml [red]), an antibody against von Willebrand factor (vWF[green]), and an antibody against glial fibrillary acidic protein (GFAP

*Figure 2 continued on next page*

Figure 2 continued

[magenta]). All three GPIHBP1-specific mAbs detected GPIHBP1 in capillaries, colocalizing with von Willebrand factor. DNA was stained with DAPI (blue). Scale bar, 50 μm. (B) Immunofluorescence confocal microscopy studies on human glioma sample 5, performed with mAbs RF4 and RE3 (10 μg/ml) in the presence or absence of 50 μg of recombinant soluble human GPIHBP1 (hGPIHBP1). Adding recombinant hGPIHBP1 to the antibody incubation abolished binding of the GPIHBP1-specific mAbs to GPIHBP1 on glioma capillaries. Images show GPIHBP1 (red), vWF (green), GFAP (magenta), and DAPI (blue). Three sections of tumors were evaluated; representative images are shown. Scale bar, 50 μm.
DOI: https://doi.org/10.7554/eLife.47178.004

specificity of the antibodies, we performed studies in which recombinant human GPIHBP1 was added to the GPIHBP1-specific mAbs before incubating the solution with the glioma sections. As expected, the presence of recombinant GPIHBP1 eliminated binding of the GPIHBP1-specific mAbs to glioma capillaries (*Figure 2B*). GPIHBP1 expression in glioma capillaries could also be detected by immunoperoxidase staining (*Figure 1—figure supplement 1*).

## GPIHBP1 is present in the capillary endothelial cells of mouse gliomas

To determine whether GPIHBP1 is expressed in a mouse model of glioblastoma, spheroids of syngeneic C57BL/6 mouse CT-2A glioma cells (*Seyfried et al., 1992*; *Oh et al., 2014*), modified to express a blue fluorescent protein (BFP) (*Mathivet et al., 2017*), were engrafted into the brains of mice harboring an endothelial cell–specific Pdgfb-iCreER$^{T2}$ transgene (*Claxton et al., 2008*) and a ROSA$^{mT/mG}$ reporter allele (*Muzumdar et al., 2007*). ROSA$^{mT/mG}$ is a two-color fluorescent, membrane-targeted Cre-dependent reporter allele. In the absence of Cre, all cells express a membrane-localized tdTomato and fluoresce red. In the setting of Cre expression, cells express membrane-localized EGFP (rather than tdTomato) and fluoresce green. Before tumor implantation, mice were injected with tamoxifen to induce Pdgfb-driven Cre expression in endothelial cells; thus, the endothelial cells of the mice expressed EGFP and fluoresced green. Mice harboring gliomas (after three weeks of growth) were injected intravenously with an Alexa Fluor 647–conjugated antibody against mouse GPIHBP1 (11A12) (*Beigneux et al., 2009*). Mice were perfused with PBS and then perfusion-fixed with 2% PFA, and tumor sections were processed for confocal immunofluorescence microscopy. GPIHBP1 was detected in endothelial cells of the gliomas, colocalizing with EGFP (brain endothelial cells), but GPIHBP1 was absent from capillaries in the adjacent normal brain (*Figure 3*, *Figure 3—figure supplement 1*). Using transmission electron microscopy, we observed large and irregularly shaped capillaries in gliomas, with numerous villus-like structures on the luminal surface of endothelial cells (*Figure 3—figure supplement 2*), similar to findings reported for capillaries in human gliomas (*Vaz et al., 1996*; *Coomber et al., 1987*; *Weller et al., 1977*).

The factors that regulate *Gpihbp1* expression in the capillary endothelial cells of peripheral tissues and gliomas are incompletely understood. However, a recent study found that *Gpihbp1* transcript levels in rat aortic endothelial cells are upregulated by vascular endothelial growth factor (VEGF) (*Chiu et al., 2016*), an angiogenic factor known to be expressed at high levels by glioma cells (*Plate et al., 1994*; *Pietsch et al., 1997*; *Christov et al., 1998*). We found that *Gpihbp1* expression in the mouse brain endothelial cell line bEnd.3 is upregulated by recombinant VEGF (*Figure 3—figure supplement 3*).

## GLUT1 is expressed in the capillaries of gliomas and normal brain

We used immunofluorescence microscopy to examine the expression of GPIHBP1 and GLUT1 (the main glucose transporter in brain capillaries [*Maher et al., 1994*; *Pardridge et al., 1990*]) in mouse gliomas and adjacent normal brain. GPIHBP1 expression was detected in gliomas but was absent in the normal brain. The signal for GLUT1 was strong in the endothelial cells of the normal brain and was easily detectable in the capillaries of gliomas (*Figure 4*, *Figure 4—figure supplements 1–2*). Consistent findings were observed in single-cell RNA-seq studies on vascular cells of gliomas (Ken Matsumoto, manuscript in preparation) and normal brain vascular cells (*Vanlandewijck et al., 2018*; *He et al., 2018d*). Endothelial cells of gliomas (identifd by high von Willebrand factor [vWF] expression) exhibit high expression of *Gpihbp1* and somewhat lower levels of *Glut1* expression (e.g., Endothelial cell cluster 5 in *Figure 4—figure supplement 3*). In normal brain, *Glut1* was highly expressed in endothelial cells, whereas *Gpihbp1* expression was absent (*Figure 4—figure*

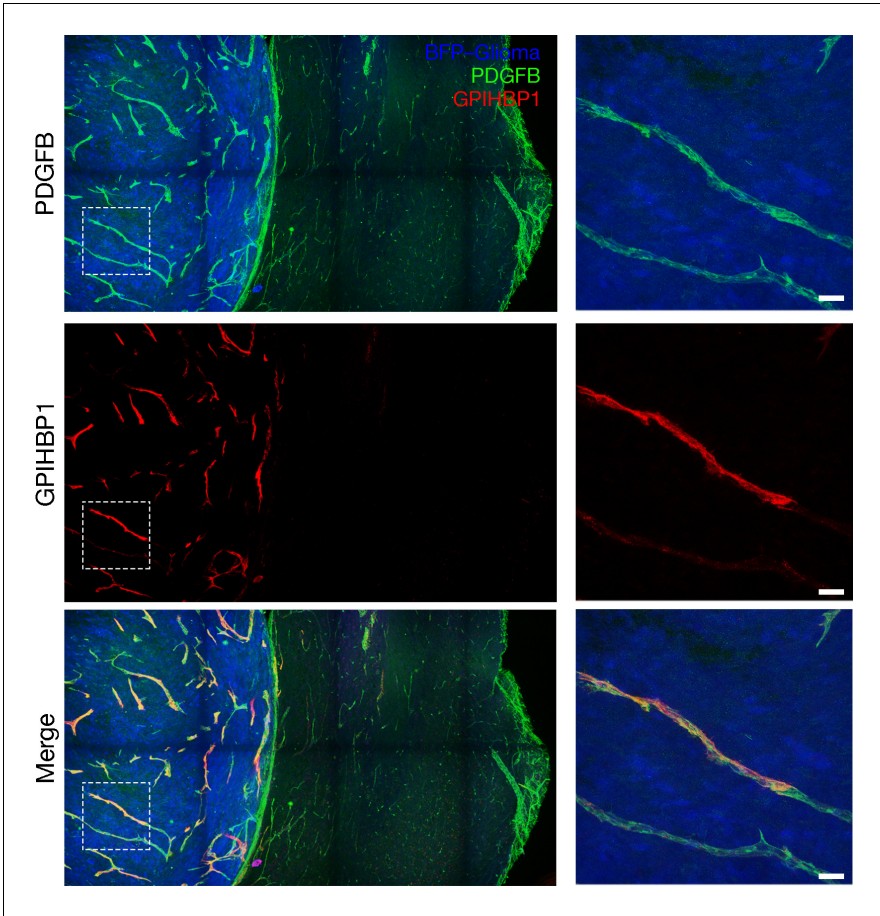

**Figure 3.** GPIHBP1 is expressed by capillary endothelial cells in mouse gliomas.  Confocal microscopy images of a BFP-tagged CT-2A glioma implanted in a ROSA^mT/mG::*Pdgfb-iCreER*^T2 mouse, revealing the expression of GPIHBP1 in capillary endothelial cells of the glioma but not those of normal brain. Tamoxifen was administered prior to implantation of the glioma spheroid to activate membrane-targeted EGFP in endothelial cells (green). After three weeks of glioma growth, mice were anesthetized and injected via the tail vein with an Alexa Fluor 647–labeled antibody against mouse GPIHBP1 (11A12; red). The mice were then perfused with PBS and perfusion-fixed with 2% PFA in PBS. Glioma and adjacent normal brain were harvested, and 200-μm-thick sections were imaged by confocal microscopy. GPIHBP1 was present on endothelial cells of the glioma (blue) but was absent from normal brain. High-magnification images of the boxed area are shown on the right. Three mice were evaluated; representative images are shown. Scale bar, 50 μm.
DOI: https://doi.org/10.7554/eLife.47178.006

The following figure supplements are available for figure 3:

**Figure supplement 1.** GPIHBP1 is expressed in the capillaries of mouse glioma but not normal brain.
DOI: https://doi.org/10.7554/eLife.47178.007

**Figure supplement 2.** The morphology of glioma capillaries differs from that of capillaries in normal brain, as revealed by transmission electron microscopy (TEM).
DOI: https://doi.org/10.7554/eLife.47178.008

**Figure supplement 3.** Vascular endothelial growth factor (VEGF) increases *Gpihbp1* transcript levels in the mouse brain microvascular endothelial cell line bEnd.3.
DOI: https://doi.org/10.7554/eLife.47178.009

*supplement 3*). In *Gpihbp1*-deficient mice, GLUT1 expression was detectable in the capillaries of gliomas and normal brain (*Figure 4—figure supplement 4*).

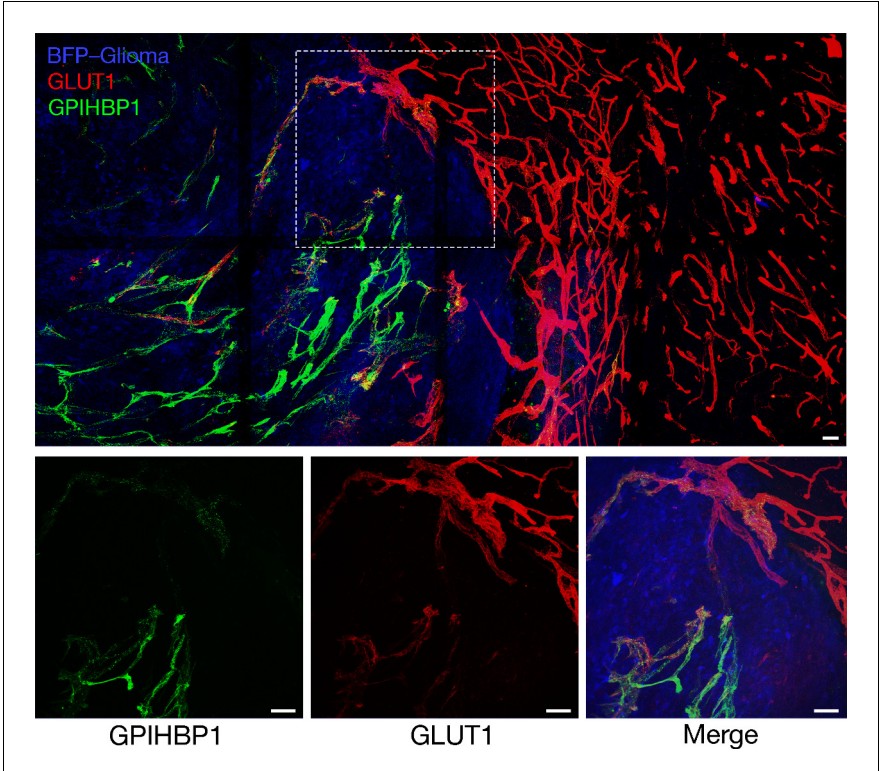

**Figure 4.** Expression of GPIHBP1 and GLUT1 in the endothelial cells of mouse gliomas. Immunohistochemical studies of a BFP-expressing CT-2A glioma (after three weeks of growth). Mice were injected *via* the tail vein with an Alexa Fluor 647–labeled antibody against mouse GPIHBP1 (11A12; green), then perfused with PBS and perfusion-fixed with 2% PFA. Glioma and adjacent normal brain tissue were harvested, then 200-μm thick sections cut, fixed with 4% PFA, and stained with an antibody against GLUT1 (red). GPIHBP1 was present in the capillaries of mouse gliomas (blue) but absent from the capillaries of the normal brain. High-magnification images in the boxed region are shown below. Three mice were evaluated; representative images are shown. Scale bar, 50 μm.
DOI: https://doi.org/10.7554/eLife.47178.010

The following figure supplements are available for figure 4:

**Figure supplement 1.** GPIHBP1 and GLUT1 expression in glioma capillaries.
DOI: https://doi.org/10.7554/eLife.47178.011

**Figure supplement 2.** GPIHBP1 and GLUT1 in glioma capillaries.
DOI: https://doi.org/10.7554/eLife.47178.012

**Figure supplement 3.** Single-cell RNA-seq observations on normal mouse brain and mouse gliomas.
DOI: https://doi.org/10.7554/eLife.47178.013

**Figure supplement 4.** GLUT1 is detectable in the endothelial cells of gliomas and normal brain in wild-type (*Gpihbp1*[+/+]) and *Gpihbp1*[−/−] mice.
DOI: https://doi.org/10.7554/eLife.47178.014

## LPL is present on GPIHBP1-expressing capillaries of mouse gliomas

Most of the LPL in peripheral tissues (e.g., heart or brown adipose tissue) is bound to GPIHBP1 on capillaries; consequently, LPL and GPIHBP1 colocalize in tissue sections (*Young et al., 2011*; *Davies et al., 2010*; *Davies et al., 2012*; *Allan et al., 2017a*; *Fong et al., 2016*; *Allan et al., 2017b*; *Allan et al., 2016*). We hypothesized that GPIHBP1-expressing endothelial cells of gliomas could capture LPL. Several observations prompted us to consider this hypothesis. First, as noted earlier, there is ample evidence for LPL expression in the brain (*Ben-Zeev et al., 1990*; *Bessesen et al., 1993*; *Goldberg et al., 1989*; *Vilaró et al., 1990*; *Yacoub et al., 1990*; *Zhang et al., 2014*), and it seemed reasonable that some of that LPL would reach high-affinity GPIHBP1-binding sites on endothelial cells. Second, gliomas contain large numbers of macrophages (F4/80-expressing cells; *Figure 5—figure supplement 1*), and macrophages are known to express LPL (*Mahoney et al., 1982*).

We found that LPL could be detected in peritoneal macrophages from wild-type mice but not in macrophages harvested from $Lpl^{-/-}$ mice carrying a skeletal muscle–specific human LPL transgene ($Lpl^{-/-}$MCK-hLPL) (*Levak-Frank et al., 1995*) (*Figure 5—figure supplement 2*). We also found that LPL could be detected in some of the macrophages in mouse gliomas and in normal brain of wild-type mice, but not in the brain of $Lpl^{-/-}$MCK-hLPL mice (*Figure 5—figure supplement 3*). These findings were consistent with single-cell RNA-seq data from glioma and normal brain, in which *Lpl* transcripts were found in the macrophages of gliomas and microglia of normal brain (*Figure 4—figure supplement 3*). *Lpl* transcripts are not present in capillary endothelial cells. Third, the most highly upregulated fatty acid metabolism gene in human gliomas, compared to normal brain tissue, is *LPL* (*Figure 5—figure supplement 4*). The second most perturbed gene in gliomas is CD36, which encodes a putative fatty acid transporter (*Figure 5—figure supplement 4*).

To determine whether LPL is bound to GPIHBP1-expressing capillaries of gliomas, we performed immunohistochemical studies, taking advantage of an affinity-purified goat antibody against mouse LPL (*Page et al., 2006*). These studies revealed colocalization of GPIHBP1 and LPL in glioma capillaries (*Figure 5*, *Figure 5—figure supplement 5*). LPL was not present in the capillaries of the normal brain or in the capillaries of gliomas from $Gpihbp1^{-/-}$ mice (*Figure 5*, *Figure 5—figure supplement 5*). As expected, the binding of the goat LPL antibody to tissues of $Lpl^{-/-}$MCK-hLPL mice was low (*Figure 5*, *Figure 5—figure supplement 5*), whereas mouse LPL was easily detectable in the heart capillaries of wild-type mice (colocalizing with GPIHBP1) (*Figure 5—figure supplement 6*). Consistent with earlier publications (*Ben-Zeev et al., 1990*; *Vilaró et al., 1990*), we observed a strong mouse LPL signal in the hippocampal neurons of wild-type mice but not of $Lpl^{-/-}$MCK-hLPL mice (*Figure 5—figure supplement 7*). Of note, LPL was undetectable in 'secondary antibody–only' experiments (i.e., when the incubation of the primary antibody with tissue sections was omitted) (*Figure 5*, *Figure 5—figure supplement 5–7*).

There is little reason to suspect that the expression of LPL influences the expression of GPIHBP1 in capillaries. The overexpression of human LPL in the skeletal muscle of $Lpl^{-/-}$MCK-hLPL mice did not alter levels of *Gpihbp1* expression (*Figure 5—figure supplement 8*).

## Margination of TRLs along glioma capillaries and uptake of TRL-derived nutrients in glioma cells

Given the presence of GPIHBP1-bound LPL on glioma capillaries, we suspected that we might find evidence of TRL margination and processing in gliomas. To test this idea, TRLs that were heavily labeled with deuterated lipids ($[^{2}H]$TRLs) (*He et al., 2018a*) were injected intravenously into mice harboring CT-2A gliomas (after three weeks of glioma growth). After allowing the $[^{2}H]$TRLs to circulate for either 1 min or 30 min, the mice were euthanized, extensively perfused with PBS, and perfusion-fixed with carbodiimide/glutaraldehyde. Heart, brain, and glioma specimens were harvested and processed for NanoSIMS imaging. $^{12}C^{14}N^{-}$ or $^{1}H^{-}$ images were used to visualize tissue morphology, and $^{2}H/^{1}H$ images were used to identify regions of $^{2}H$ enrichment. The scale in the $^{2}H/^{1}H$ images of brain and glioma specimens ranges from 0.00018 to 0.0003 (i.e., from levels slightly above $^{2}H$ natural abundance to levels twice as high as $^{2}H$ natural abundance). The scale in the heart $^{2}H/^{1}H$ images ranges from 0.00018 to 0.0006. In mice euthanized 1 min after the $[^{2}H]$TRLs injection, $[^{2}H]$TRL margination was visualized along the luminal surface of glioma and heart capillaries, but not along the capillaries of normal brain (*Figure 6A–B*). After 1 min, deuterated lipids from the $[^{2}H]$TRLs had already entered glioma cells and were even found in cytosolic neutral lipid droplets of those cells (*Figure 6B*). By contrast, $^{2}H$ enrichment was virtually absent in normal brain. As expected (*He et al., 2018a*), we observed substantial amounts of $[^{2}H]$TRL-derived lipids in cardiomyocytes, including in cytosolic lipid droplets. In gliomas harvested 30 min after the injection of $[^{2}H]$TRLs, we observed similar findings: TRL margination along capillaries of gliomas and heart and the uptake of TRL-derived nutrients by glioma cells and cardiomyocytes (*Figure 7*). Again, $[^{2}H]$TRL margination was absent in capillaries of the normal brain at the 30-min time point, and we did not find $^{2}H$ enrichment in the parenchymal cells of the normal brain. We did, however, observe very low levels of $^{2}H$ enrichment in capillary endothelial cells of normal brain. Given the absence of TRL margination in normal brain capillaries, we speculate that the very low amounts of $^{2}H$ enrichment in brain capillary endothelial cells may relate to $[^{2}H]$TRL processing in the periphery, followed by the uptake of unesterified $[^{2}H]$fatty acids by endothelial cells of the brain.

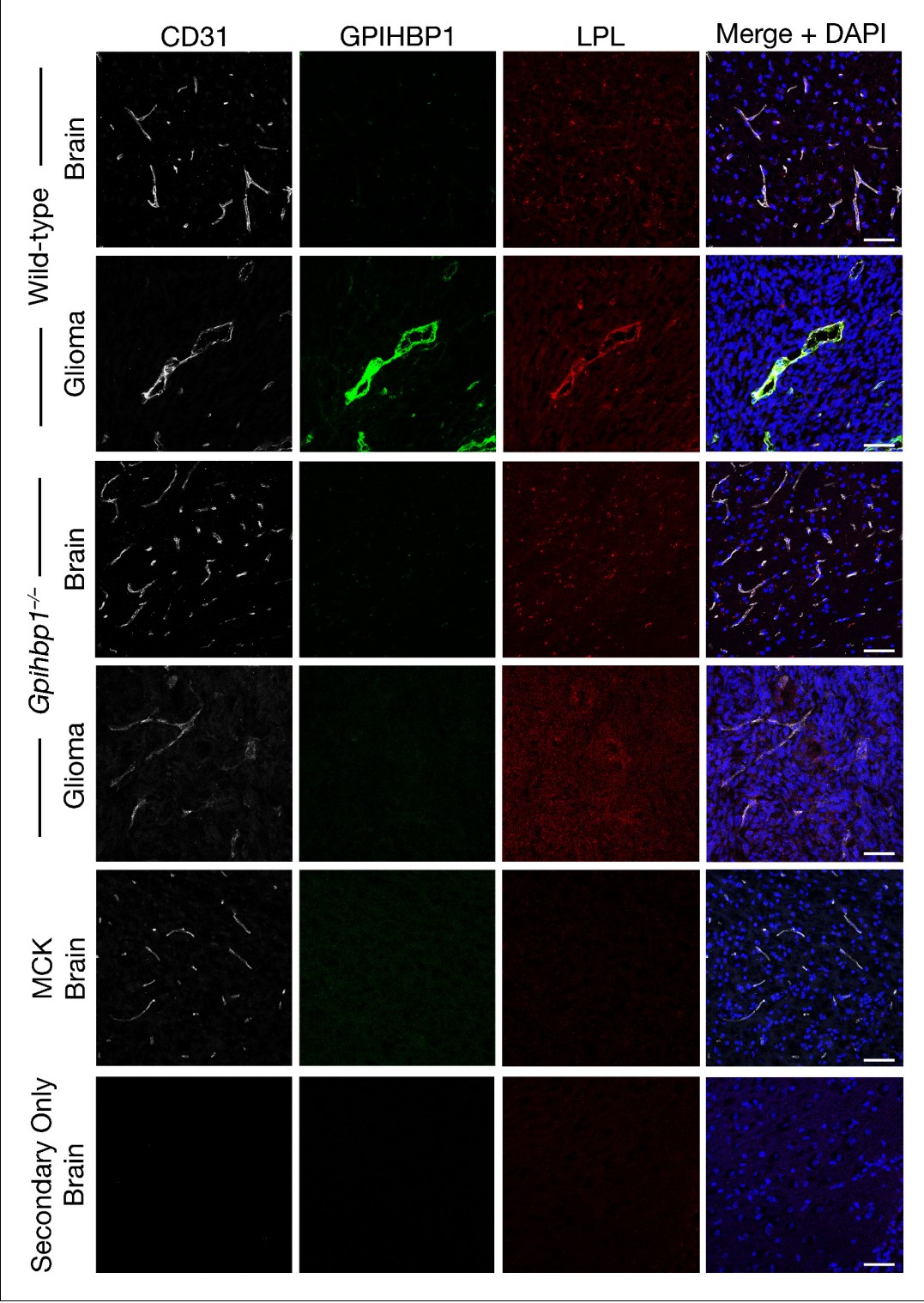

**Figure 5.** Lipoprotein lipase (LPL) colocalizes with GPIHBP1 in glioma capillaries. Confocal immunofluorescence microscopy studies on glioma and normal brain from wild-type and *Gpihbp1*$^{-/-}$ mice, along with the brain from an *Lpl*$^{-/-}$ mouse carrying a skeletal muscle–specific human LPL transgene (MCK). Glioma and brain sections (10-μm-thick) were fixed with 3% PFA and then stained with a mAb against mouse GPIHBP1 (11A12; green), a goat antibody against mouse LPL (red), and a rabbit antibody against CD31 (white). LPL colocalizes with GPIHBP1 and CD31 in the capillaries of gliomas; GPIHBP1 and LPL were absent from normal brain capillaries and from glioma capillaries in *Gpihbp1*$^{-/-}$ mice. DNA was stained with DAPI (blue). No LPL was detected in the capillaries of *Lpl-*
*Figure 5 continued on next page*

*Figure 5 continued*

deficient mice (MCK) or when the incubation with primary antibodies was omitted (Secondary Only). Staining of all tissue sections was performed simultaneously, and all images were recorded with identical microscopy settings. Three mice per genotype were evaluated; representative images are shown. Scale bar, 50 μm.

DOI: https://doi.org/10.7554/eLife.47178.015

The following figure supplements are available for figure 5:

**Figure supplement 1.** Large numbers of macrophages in mouse gliomas.

DOI: https://doi.org/10.7554/eLife.47178.016

**Figure supplement 2.** LPL is expressed in peritoneal macrophages from wild-type mice but not in macrophages from *Lpl*$^{-/-}$MCK-hLPL mice, as revealed by confocal immunofluorescence microscopy.

DOI: https://doi.org/10.7554/eLife.47178.017

**Figure supplement 3.** LPL is present in the macrophages of brain and gliomas, as revealed by confocal immunofluorescence microscopy.

DOI: https://doi.org/10.7554/eLife.47178.018

**Figure supplement 4.** Heat map showing genes related to fatty acid metabolism that are upregulated in human gliomas, compared to normal brain.

DOI: https://doi.org/10.7554/eLife.47178.019

**Figure supplement 5.** LPL colocalizes with GPIHBP1 in glioma capillaries, as revealed by confocal immunofluorescence microscopy.

DOI: https://doi.org/10.7554/eLife.47178.020

**Figure supplement 6.** Mouse LPL is absent from tissues of an *Lpl*$^{-/-}$MCK-hLPL mouse, as revealed by confocal immunofluorescence microscopy.

DOI: https://doi.org/10.7554/eLife.47178.021

**Figure supplement 7.** LPL is present in the hippocampal neurons of wild-type mice, as revealed by confocal immunofluorescence microscopy.

DOI: https://doi.org/10.7554/eLife.47178.022

**Figure supplement 8.** Mouse *Gpihbp1*, mouse *Lpl,* and human *LPL* transcript levels in 25-week-old wild-type and *Lpl*$^{-/-}$MCK-hLPL mice (MCK-hLPL).

DOI: https://doi.org/10.7554/eLife.47178.023

At both the 1-min and 30-min time points, we observed heterogeneity in $^2$H enrichment in glioma cells, with occasional perivascular cells exhibiting striking $^2$H enrichment. We do not know the identity of the highly enriched perivascular cells (i.e., whether they are tumor cells, pericytes, or macrophages), nor do we understand why some cells within the glioma took up more [$^2$H]TRL-derived lipids than other cells.

As an experimental control, we injected a mouse with PBS alone rather than with [$^2$H]TRLs. As expected, there was no $^2$H enrichment in the tissues of that mouse (*Figure 7—figure supplement 1*).

We performed an additional study in which [$^2$H]TRLs were injected intravenously into a wild-type mouse and a *Gpihbp1*$^{-/-}$ mouse. After 15 min, the hearts and brains from these mice were harvested and processed for NanoSIMS imaging. The $^2$H/$^1$H ratio images revealed $^2$H enrichment in the heart of the wild-type mouse but negligible $^2$H enrichment in the heart of the *Gpihbp1*$^{-/-}$ mouse ($^2$H enrichment in cardiomyocyte lipid droplets was only ~10% greater than natural abundance) (*Figure 7—figure supplement 2*). In hindsight, the negligible amounts of $^2$H enrichment in the heart of the *Gpihbp1*$^{-/-}$ mouse was probably not surprising, given the very large pool of unlabeled triglycerides in the bloodstream of *Gpihbp1*$^{-/-}$ mice (~50–100-fold higher than that in wild-type mice). At the 15-min time point, we were unable to detect $^2$H enrichment in the brain of either the wild-type mouse or the *Gpihbp1*$^{-/-}$ mouse (*Figure 7—figure supplement 2*).

## $^{13}$C enrichment in gliomas following administration of $^{13}$C-labeled fatty acids or $^{13}$C-labeled glucose by gastric gavage

In addition to studies of gliomas after an intravenous injection of [$^2$H]TRLs, we performed NanoSIMS imaging after administering $^{13}$C-labeled fatty acids or $^{13}$C-labeled glucose by gastric gavage (three doses over 36 hr) (*Figure 8*). In the case of the $^{13}$C-labeled fatty acid experiments, it is likely that most of the $^{13}$C-labeled lipids entered the bloodstream in chylomicrons. Once again, $^{12}$C$^{14}$N$^-$ images were useful for tissue morphology, and the $^{13}$C/$^{12}$C ratio images were useful to identify

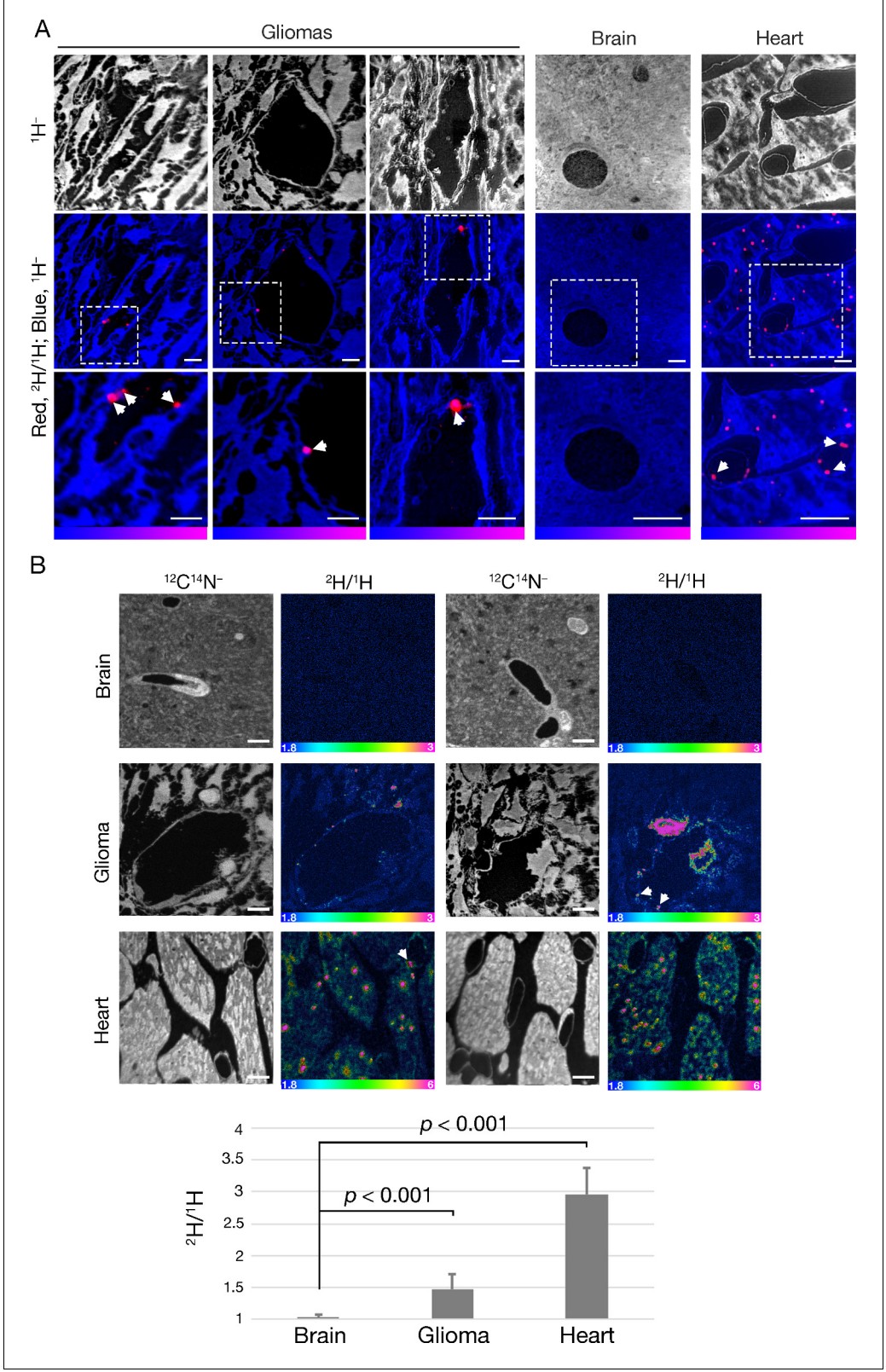

**Figure 6.** NanoSIMS imaging reveals margination of [²H]TRLs along glioma capillaries and ²H enrichment in adjacent glioma cells. Four-month-old C57BL/6 mice harboring CT-2A gliomas were fasted for 4 hr and then injected intravenously with 200 μl of [²H]TRLs. After 1 min, mice were euthanized and perfusion-fixed with carbodiimide/glutaraldehyde. Tissue sections were processed for NanoSIMS imaging. (**A**) NanoSIMS images showing margination of [²H]TRLs in glioma capillaries. ¹H⁻ images were created to visualize tissue morphology (upper panels). Composite ²H/¹H (red)

*Figure 6 continued on next page*

Figure 6 continued

and $^1H^-$ (blue) images reveal [$^2H$]TRLs (white arrows) in glioma and heart capillaries (middle and lower panels). The lower panels are close-up images of the regions outlined in the middle panels. $^2H/^1H$ ratio scales were set to show marginated TRLs. Scale bars, 4 μm. (**B**) NanoSIMS images showing $^2H$ enrichment in glioma tissue. $^{12}C^{14}N^-$ images were generated to visualize tissue morphology. $^2H/^1H$ ratio images reveal margination of [$^2H$]TRLs within the capillary lumen and $^2H$-enriched lipid droplets in gliomas and heart. There was no $^2H$ enrichment in normal brain tissue. Scale bars, 4 μm. The bar graph shows the average fold change ± SD in the $^2H/^1H$ ratio above natural abundance. The experiment was performed in two mice with a minimum of seven images analyzed for each sample. Differences were assessed using a Student's *t*-test with Welch's correction.

DOI: https://doi.org/10.7554/eLife.47178.024

regions of $^{13}C$ enrichment. The scale for the $^{13}C/^{12}C$ images ranges from 0.0115 to 0.0150 (from slightly above $^{13}C$ natural abundance to ~36% greater than natural abundance). After administering $^{13}C$-labeled fatty acids, $^{13}C$ enrichment was observed in both glioma cells and in the capillary endothelial cells of gliomas (*Figure 8A*). In some images, $^{13}C$-enriched cytosolic lipid droplets were visible in glioma cells (*Figure 8—figure supplement 1*). $^{13}C$ enrichment was virtually absent from normal brain (*Figure 8A*). However, after adjusting the scale of the NanoSIMS images, a small amount of $^{13}C$ enrichment was observed in capillary endothelial cells within the brain parenchyma (*Figure 8—figure supplement 2*). As expected (*He et al., 2018a*), we observed substantial amounts of $^{13}C$ enrichment in cardiomyocytes (*Figure 8A*).

After administering [$^{13}C$]glucose to mice, $^{13}C$ enrichment was easily detectable in normal brain but was even ~20% higher in gliomas (*Figure 8B*). We also observed $^{13}C$ enrichment in cardiomyocytes (*Figure 8B*). As expected, there was no $^{13}C$ enrichment in the tissues of a mouse that was administered PBS alone (*Figure 8—figure supplement 3*).

To determine whether an absence of GPIHBP1 expression would influence the growth of glioma tumors, CT-2A glioma cells that had been stably transfected with a *Gaussia* luciferase reporter were injected into the brains of wild-type and *Gpihbp1$^{-/-}$* mice (n = 11/group). Tumor burden was assessed in live animals by measuring luciferase activity in the blood (*Mai et al., 2017*; *Tannous, 2009*). We observed no statistically significant differences in tumor growth, tumor size, or survival between wild-type and *Gpihbp1$^{-/-}$* mice (*Figure 8—figure supplement 4*). This result was not particularly surprising, given that gliomas have a robust capacity to utilize glucose-derived nutrients (*Figure 8B*).

## Discussion

We sought to determine whether GPIHBP1, despite its complete absence from the capillaries of the brain, might nevertheless be expressed in the capillaries of gliomas. Using standard immunohistochemistry procedures, we documented GPIHBP1 expression in capillary endothelial cells of human gliomas and CT-2A-derived mouse gliomas. The expression of GPIHBP1 in glioma capillaries was intriguing, but the crucial issue is whether LPL would be bound to the GPIHBP1. Additional immunohistochemistry studies on mouse gliomas revealed that LPL colocalizes with GPIHBP1 on glioma capillaries, just as LPL colocalizes with GPIHBP1 in the capillaries of heart and brown adipose tissue (*Young et al., 2011*; *Davies et al., 2010*; *Davies et al., 2012*; *Allan et al., 2017a*; *Fong et al., 2016*; *Allan et al., 2017b*; *Allan et al., 2016*). The binding of LPL to GPIHBP1 was specific: the LPL-specific goat antibody did not detect LPL in the capillaries of gliomas in *Gpihbp1$^{-/-}$* mice, nor did it detect any LPL in macrophages or hippocampal neurons of *Lpl$^{-/-}$*MCK-hLPL mice. The colocalization of GPIHBP1 and LPL in the capillaries of gliomas implied that we might find evidence for TRL margination and processing in these tumors. Indeed, we observed both [$^2H$]TRL margination along glioma capillaries and the entry of TRL-derived nutrients into glioma cells. Consistent with results of earlier studies (*Goulbourne et al., 2014*; *He et al., 2018a*), TRL margination was absent from the capillaries of normal brain, and we found no $^2H$ enrichment in the brain parenchyma. We did, however, find very low levels of $^2H$ enrichment in capillary endothelial cells of normal brain, perhaps as a result of the uptake of fatty acids that are derived from TRL processing in peripheral tissues. We observed consistent findings after administering [$^{13}C$]fatty acids to mice by gastric gavage. In those experiments, we observed strong $^{13}C$ enrichment in gliomas but no $^{13}C$ enrichment in the normal brain (except for low levels of enrichment in capillary endothelial cells). After administering [$^{13}C$]glucose by gavage, $^{13}C$ enrichment was observed in both gliomas and normal brain. It is important to

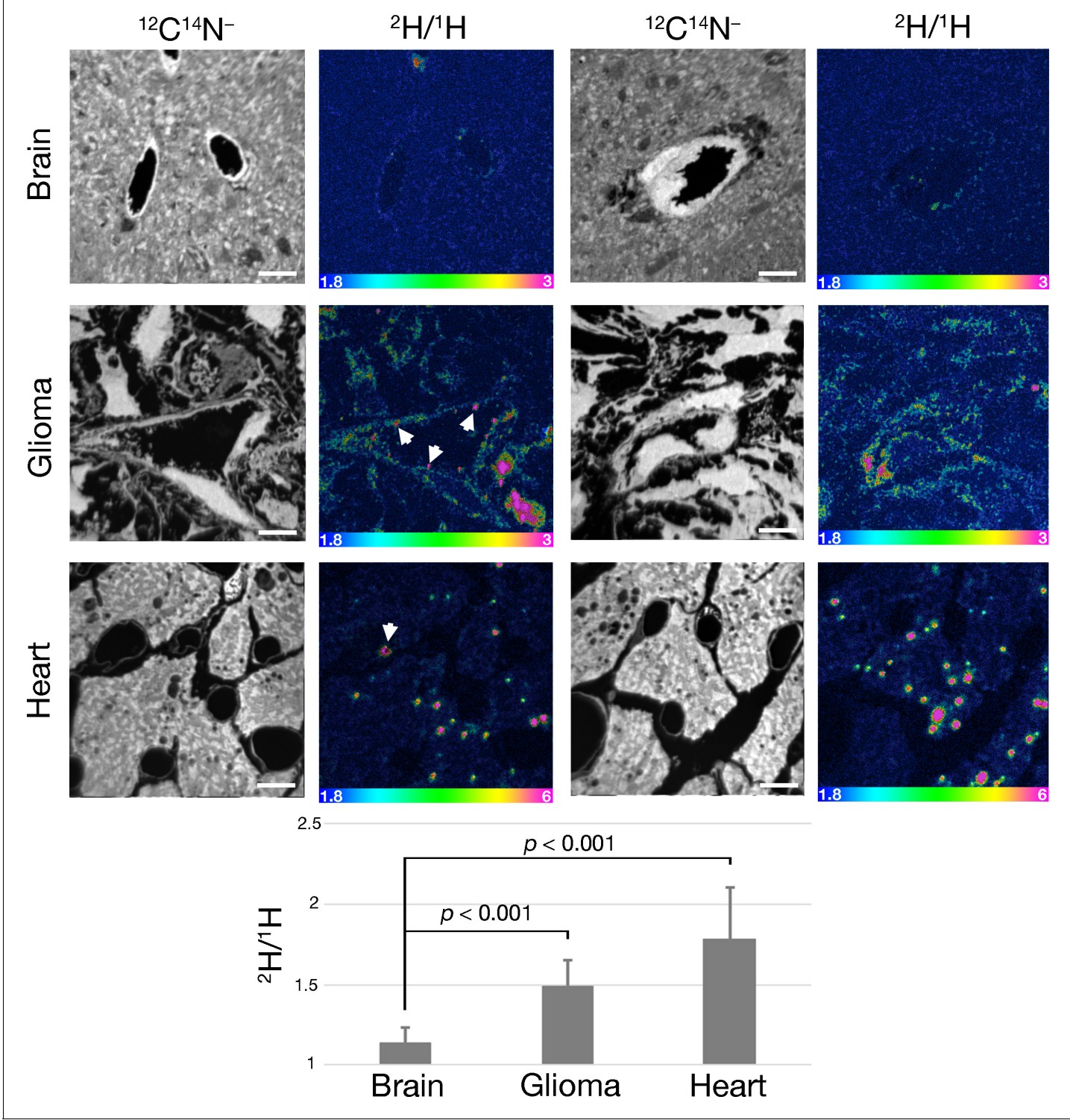

**Figure 7.** NanoSIMS imaging showing $^2$H enrichment in gliomas 30 min after an intravenous injection of [$^2$H]TRLs. Four-month-old C57BL/6 mice harboring CT-2A gliomas were fasted for 4 hr and then injected intravenously with 200 μl of [$^2$H]TRLs. After 30 min, mice were euthanized and perfusion-fixed with carbodiimide/glutaraldehyde. Sections of glioma, brain, and heart were processed for NanoSIMS imaging. $^{12}$C$^{14}$N$^-$ images were created to visualize tissue morphology. $^2$H/$^1$H ratio images reveal margination of [$^2$H]TRLs along the capillary lumen (white arrows) and $^2$H enrichment in glioma and heart, including in cytosolic lipid droplets. Images of normal brain revealed slight $^2$H enrichment in capillary endothelial cells. Scale bars, 4 μm. The bar graph shows the average fold change ± SD in the $^2$H/$^1$H ratio above natural abundance. The experiment was performed in two mice, with a minimum of seven images analyzed for each sample. Differences were assessed with a Student's $t$-test with Welch's correction.

*Figure 7 continued on next page*

*Figure 7 continued*

DOI: https://doi.org/10.7554/eLife.47178.025

The following figure supplements are available for figure 7:

**Figure supplement 1.** Absence of $^2$H enrichment in glioma and brain of a mouse that had been given an injection of PBS alone.

DOI: https://doi.org/10.7554/eLife.47178.026

**Figure supplement 2.** Uptake of [$^2$H]TRLs in heart and brain of *Gpihbp1$^{-/-}$* mice.

DOI: https://doi.org/10.7554/eLife.47178.027

note that the [$^{13}$C]fatty acids and the [$^{13}$C]glucose were administered in three doses over 36 hr before harvesting tissues for NanoSIMS analyses, allowing ample time for the labeled nutrients to be utilized as fuel or to be converted into other nutrients (e.g., nonessential amino acids) (*He et al., 2018a*; *Sidossis et al., 1995*; *Schneider and Potter, 1957*). Thus, after administering $^{13}$C-labeled

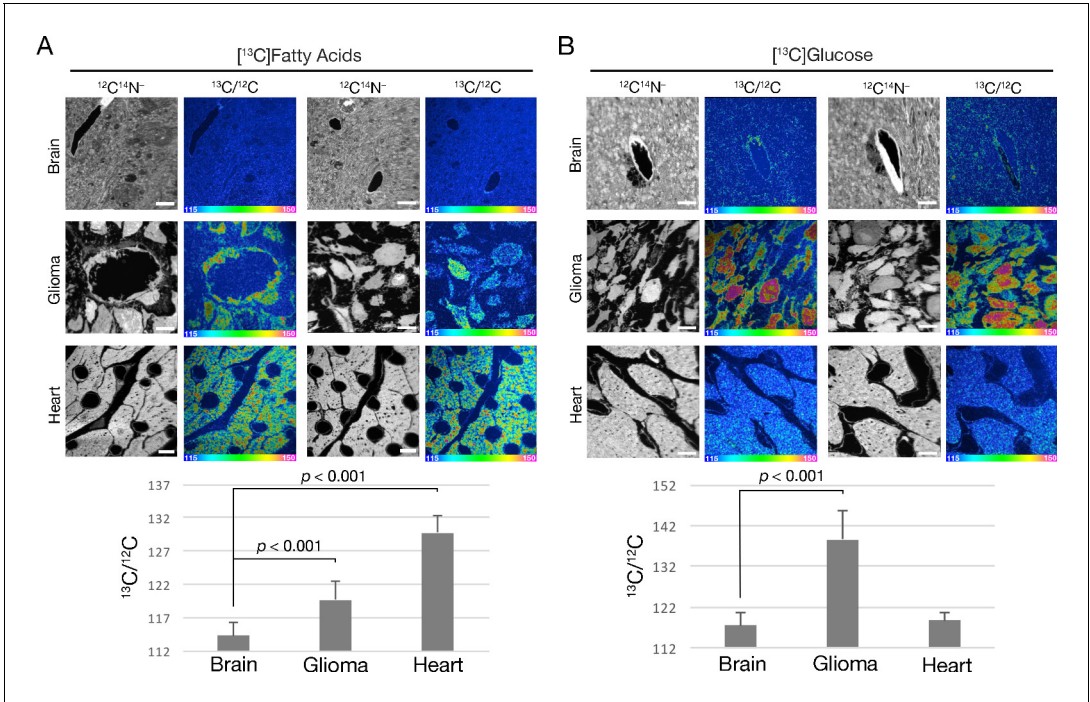

**Figure 8.** Tissue uptake of fatty acid and glucose-derived nutrients by mice harboring CT-2A gliomas. (**A**) NanoSIMS images showing $^{13}$C enrichment in mouse tissues (brain, glioma, and heart) after oral administration of $^{13}$C-labeled mixed fatty acids to mice (three 80-mg doses administered 12 h apart). $^{12}$C$^{14}$N$^-$ images were generated to visualize tissue morphology; $^{13}$C/$^{12}$C ratio images were used to visualize $^{13}$C enrichment in tissues. Scale bars, 4 μm. (**B**) NanoSIMS images revealing $^{13}$C enrichment in tissues following oral administration of $^{13}$C-labeled glucose to mice (three 75-mg doses given 12-h apart). $^{12}$C$^{14}$N$^-$ images were generated to visualize tissue morphology; $^{13}$C/$^{12}$C ratio images were generated to assess $^{13}$C enrichment in tissues. Scale bars, 4 μm. The bar graphs show the average $^{13}$C/$^{12}$C ratio ± SD multiplied by 10,000 for fatty acids (left) and glucose (right). Each experiment was performed in two mice, with a minimum of seven images analyzed for each sample. Differences were assessed using a Student's *t*-test with Welch's correction.

DOI: https://doi.org/10.7554/eLife.47178.028

The following figure supplements are available for figure 8:

**Figure supplement 1.** $^{13}$C-enriched lipid droplets in mouse glioma cells.

DOI: https://doi.org/10.7554/eLife.47178.029

**Figure supplement 2.** NanoSIMS imaging showing $^{13}$C enrichment in capillary endothelial cells of normal brain after administering $^{13}$C-labeled mixed fatty acids by oral gavage (three doses of 80 mg administered 12 hr apart).

DOI: https://doi.org/10.7554/eLife.47178.030

**Figure supplement 3.** Absence of $^{13}$C enrichment in the glioma and brain of a mouse that had been given an injection of PBS alone.

DOI: https://doi.org/10.7554/eLife.47178.031

**Figure supplement 4.** Glioma studies in *Gpihbp1$^{+/+}$* and *Gpihbp1$^{-/-}$* mice.

DOI: https://doi.org/10.7554/eLife.47178.032

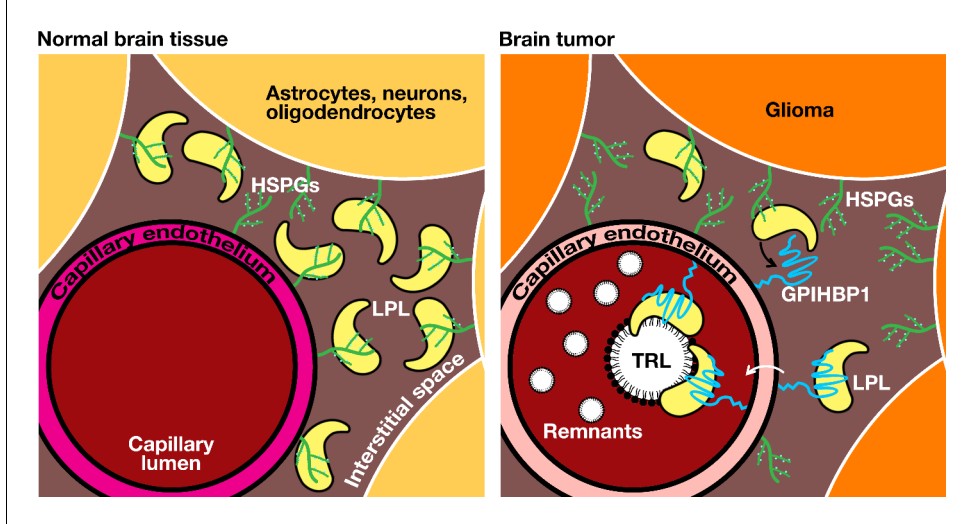

**Figure 9.** Intravascular lipolysis as a source of lipid nutrients for gliomas. In normal brain (left panel), LPL is produced by astrocytes, neurons, oligodendrocytes, and fibroblasts. Because GPIHBP1 is not expressed in the capillaries of the brain parenchyma, we have proposed that LPL remains within the interstitial spaces of the brain (i.e., that it has an extravascular function) (*Adeyo et al., 2012*; *Young et al., 2011*). In gliomas (right panel), GPIHBP1 is expressed in capillary endothelial cells, allowing GPIHBP1 to capture locally produced LPL and to shuttle it to the capillary lumen. Intravascular processing of triglyceride-rich lipoproteins in gliomas provides a source of lipid nutrients for glioma cells. HSPGs, heparan sulfate proteoglycans.
DOI: https://doi.org/10.7554/eLife.47178.033

fatty acids or glucose to mice, the [13]C in glioma cells was probably present in a variety of macromolecules (e.g., glucose, lipids, proteins, and nucleic acids).

Documenting GPIHBP1 and LPL in glioma capillaries, combined with the discovery that TRL-derived nutrients are taken up and utilized by glioma cells, opens a new chapter in glioma metabolism research (*Figure 9*). Laboratories that are interested in glioma metabolism have typically focused on the intrinsic metabolic properties of glioma cells and on how metabolic pathways in gliomas differ from those in normal brain (*Lin et al., 2017*; *Guo et al., 2009a*; *Guo et al., 2009b*; *Guo et al., 2013*; *Gopal et al., 1963*; *Strickland and Stoll, 2017*; *Agnihotri and Zadeh, 2016*). There have been suggestions, based on indirect observations of substrate utilization, that glioma tumors are capable of utilizing fatty acids for fuel and for anabolic processes (*Lin et al., 2017*; *Guo et al., 2013*; *Mashimo et al., 2014*; *Ru et al., 2013*; *Zaidi et al., 2013*). In those studies, however, the assumption was that the fatty acids probably originated from the tumor cells by *de novo* lipogenesis (*Guo et al., 2011*; *Guo et al., 2009a*; *Guo et al., 2009b*). No one, as far as we are aware, had ever considered the possibility that gliomas might be capable of taking up and utilizing nutrients from LPL-mediated intravascular processing of TRLs.

In an ultrastructural study of human gliomas, *Vaz et al. (1996)* commented that the morphology of endothelial cells in gliomas resembles that of capillary endothelial cells in peripheral tissues, with euchromatin-rich nuclei, occasional fenestrations, and numerous pinocytotic vesicles within the cytoplasm. The expression of GPIHBP1 (a hallmark of capillary endothelial cells in peripheral tissues) in gliomas provides biochemical support for the notion that glioma capillaries resemble capillaries in peripheral tissues (*Vaz et al., 1996*). Our electron microscopy studies confirmed that the morphological features of glioma capillaries and normal brain capillaries differ substantially.

We have relatively few insights into the molecular basis for GPIHBP1 expression in glioma capillaries. One possibility is that the absence of a blood–brain barrier in glioma capillaries (*Dubois et al., 2014*; *Wolburg et al., 2012*; *Liebner et al., 2000*; *Sage and Wilson, 1994*) permits the exposure of endothelial cells to a paracrine factor that activates GPIHBP1 expression. Another possibility is that GPIHBP1 expression is stimulated by the expression of VEGF that is produced by glioma cells (*Plate et al., 1994*; *Pietsch et al., 1997*; *Christov et al., 1998*). In our studies, VEGF increased GPIHBP1 expression in the mouse brain endothelial cell line bEnd.3.

In the past, other laboratories have reported that glioma tumor cells can transdifferentiate into endothelial cells, thereby augmenting the vascular supply to tumors (*Wang et al., 2010*; *Ricci-Vitiani et al., 2010*; *Soda et al., 2011*). For example, endothelial cells in human glioblastomas were reported to harbor the same genetic alterations as the tumor cells, implying that at least some of the glioblastoma endothelial cells originate from stem cells within the tumor (*Wang et al., 2010*; *Ricci-Vitiani et al., 2010*). In another model (*Soda et al., 2011*), a Cre recombinase (*Cre*)-loxP–controlled lentiviral vector encoding activated forms of H-Ras and Akt was injected into the hippocampus of GFAP-*Cre* p53 mice, eliciting glioblastomas. In that model, the oncogenes were expressed in the GFAP+ cells, and the resulting tumors expressed GFP, H-Ras, and Akt and the loss of p53. Some GFP+ endothelial cells were observed in tumors, implying that these endothelial cells had originated from tumor cells. Furthermore, implanting a tumor cell line (generated from tumors induced with the same lentiviral vector) into the brain of immunocompromised mice was reported to yield tumors containing GFP+ endothelial cells. In our current studies, we observed no evidence of the differentiation of glioma cells into capillary endothelial cells. The glioma cell line that we used expressed blue fluorescent protein (BFP), but we did not find BFP expression in the capillary endothelial cells of gliomas.

Mass spectrometry–based analyses of homogenized tissue extracts from mouse gliomas and normal brain tissue, along with similar analyses of tumors from human patients, suggested differences in acetate oxidation in gliomas *vs.* normal brain (*Mashimo et al., 2014*). Although these studies of tissue extracts have been useful, they obviously cannot provide anatomical insights into metabolism. We have argued that NanoSIMS imaging studies are particularly useful when the goal is to understand metabolism at an anatomic level (cellular or subcellular) (*He et al., 2018a*). In the current studies, NanoSIMS imaging provided anatomic insights into glioma metabolism. For example, we observed TRL margination along the capillaries of gliomas but not in the capillaries of adjacent normal brain tissue. We also showed that the transport of TRL-derived nutrients across glioma capillaries and into glioma cells is rapid, occurring within 1 min, and that there is heterogeneity in nutrient uptake by different cells within the tumor. We found no uptake of TRL-derived nutrients by normal brain 1 or 15 min after the injection of [$^2$H]TRLs and only very small amounts (confined to capillary endothelial cells) after 30 min. In addition, following the administration of [$^{13}$C]glucose, we found more $^{13}$C enrichment in gliomas than in normal brain. As far as we are aware, our study is the first to use NanoSIMS analyses to investigate cancer metabolism *in vivo*. As we look to the future, we have little doubt that NanoSIMS imaging will be an important tool for understanding tumor metabolism, making it possible to investigate metabolic heterogeneity in tumor cells along with the metabolic properties of vascular cells, fibroblasts, and macrophages within the tumor. Nevertheless, it is important to point out that NanoSIMS imaging is not high-throughput, at least with the current instruments, and for that reason NanoSIMS imaging is best used (as in this study) to address discrete anatomic issues in metabolism. Examining large numbers of tumors or large numbers of mice would be difficult. Also, NanoSIMS imaging is very expensive.

Our studies have provided fresh insights into the uptake of lipid nutrients by gliomas, but many issues remain to be investigated. For example, in the current studies, we found numerous macrophages within gliomas, but we did not address differences in nutrient uptake by macrophages and glioma cells. In future studies, it should be possible to examine the uptake of TRL-derived nutrients into tumor cells, macrophages, and other immune cells within gliomas (by identifying specific cell types with $^{15}$N-labeled monoclonal antibodies or antibodies tagged with different lanthanide metals [*Waentig et al., 2012*; *Kanje et al., 2016*; *Angelo et al., 2014*; *Keren et al., 2018*]). It would also be desirable to determine whether the uptake of TRL-derived nutrients in gliomas correlates with the levels of GPIHBP1 and LPL in glioma capillaries (as quantified with LPL- and GPIHBP1-specific antibodies tagged with different lanthanide metals). Finally, it would be desirable to investigate whether the presence of GPIHBP1 and LPL in glioma capillaries could be exploited for patient care. For example, it is conceivable that fluorescently labeled GPIHBP1 antibodies or DiI-labeled TRLs could guide the surgical resection of tumors. In addition, a localized injection of GPIHBP1-specific monoclonal antibodies conjugated to chemotherapeutic agents into gliomas might be useful in targeting tumor vasculature (*Schrama et al., 2006*). A localized injection of gold-conjugated GPIHBP1-specific monoclonal antibodies could augment the efficacy of external beam radiotherapy (*Haume et al., 2016*; *Hainfeld et al., 2004*; *Hainfeld et al., 2008*).

# Materials and methods

## Key resources table

| Reagent type (species) or resource | Designation | Source or reference | Identifiers | Additional information |
|---|---|---|---|---|
| Genetic reagent (*M. musculus*) | *Gpihbp1*$^{-/-}$ | *Beigneux et al., 2007* | RRID: MGI:3771172 | Dr. Stephen G Young (UCLA) |
| Genetic reagent (*M. musculus*) | *Lpl*$^{-/-}$MCK-hLPL | *Levak-Frank et al., 1995* | RRID: MGI:3624988 | Dr. Rudolph Zechner (Graz University) |
| Genetic reagent (*M. musculus*) | ROSA$^{mT/mG}$ Pdgfb-iCre$^{T2}$ | *Mathivet et al., 2017* | | Dr. Holger Gerhardt (VIB KU-Leuven) |
| Cell line (*M. musculus*) | CT-2A | *Seyfried et al., 1992* | | Dr. Thomas Seyfried (Boston College) |
| Cell line (*M. musculus*) | CT-2A–BFP | PMID: 24658686 | | Dr. Holger Gerhardt (VIB KU-Leuven) |
| Cell line (*M. musculus*) | bEnd.3 | ATCC | Catalog No. CRL-2299 RRID: CVCL_0170 | |
| Transfected construct (lentiviral plasmid) | plenti-GLuc-IRES-EGFP | Targeting Systems | Catalog No. GL-GFP | |
| Antibody | Rat monoclonal anti-mouse GPIHBP1 (11A12) | *Beigneux et al., 2009* | | Dr. Stephen G Young (UCLA); IHC (10 µg/ml) |
| Antibody | Mouse monoclonal anti-human GPIHBP1 (RE3) | *Hu et al., 2017* | | Dr. Stephen G Young (UCLA); IHC (10 µg/ml) |
| Antibody | Mouse monoclonal anti-human GPIHBP1 (RF4) | *Hu et al., 2017* | | Dr. Stephen G Young (UCLA); IHC (10 µg/ml) |
| Antibody | Mouse monoclonal anti-human GPIHBP1 (RG3) | *Hu et al., 2017* | | Dr. Stephen G Young (UCLA); IHC (10 µg/ml) |
| Antibody | Rabbit polyclonal anti-vWF | Dako | Catalog No. A0082 RRID: AB_2315602 | IHC (1:200) |
| Antibody | Goat polyclonal anti-GFAP | Abcam | Catalog No. ab53554 RRID: AB_880202 | IHC (1:200) |
| Antibody | Rabbit polyclonal anti-GLUT1 | Millipore-Sigma | Catalog No. 07–1401 RRID: AB_1587074 | IHC (1:200) |
| Antibody | Rabbit polyclonal anti-CD31 | Abcam | Catalog No. ab28364 RRID: AB_726362 | IHC (1:50) |
| Antibody | Rat monoclonal anti-F4/80 | Abcam | Catalog No. ab6640 RRID: AB_1140040 | IHC (10 µg/ml) |
| Antibody | Goat polyclonal anti-mouse LPL | *Page et al., 2006* | | Dr. André Bensadoun (Cornell); IHC (12 µg/ml) |
| Antibody | Alexa Fluor 488, 568, 647 secondaries | ThermoFisher Scientific | | IHC (1:500) |
| Commercial assay or kit | ImmPRESS Excel Staining Kit | Vector Laboratory | Catalog No. MP-7602 | |
| Sequence-based reagent | Mouse *Gpihbp1* primers | | | 5′-AGCAGGGACAGAGCACCTCT-3′ and 5′-AGACGAGCGTGATGCAGAAG-3′ |
| Sequence-based reagent | Mouse *Cd31* primers | | | 5′-AACCGTATCTCCAAAGCCAGT-3′ and 5′-CCAGACGACTGGAGGAGAACT-3′ |
| Sequence-based reagent | Mouse *Angpt2* primers | | | 5′-AACTCGCTCCTTCAGAAGCAGC-3′ and 5′-TTCCGCACAGTCTCTGAAGGTG-3′ |

*Continued on next page*

*Continued*

| Reagent type (species) or resource | Designation | Source or reference | Identifiers | Additional information |
|---|---|---|---|---|
| Sequence-based reagent | Mouse *Dusp5* primers | | | 5′-TCGCCTACAGACCAGCCTATGA-3′ and 5′-TGATGTGCAGGTTGGCGAGGAA-3′ |
| Sequence-based reagent | Mouse *Cxcr4* primers | | | 5′-GACTGGCATAGTCGGCAATGGA-3′ and 5′-CAAAGAGGAGGTCAGCCACTGA-3′ |
| Sequence-based reagent | Mouse *Lpl* primers | | | 5′-AGGTGGACATCGGAGAACTG-3′ and 5′-TCCCTAGCACAGAAGATGACC-3′ |
| Sequence-based reagent | Human *LPL* primers | | | 5′-TAGCTGGTCAGACTGGTGGA-3′ and 5′-TTCACAAATACCGCAGGTG-3′ |
| Recombinant DNA reagent | ALO-D4 plasmid | *Gay et al., 2015* | | Dr. Arun Radhakrishnan (UT Southwestern) |
| Chemical compound, drug | N-(3-Dimethylaminopropyl)-N′-ethylcarbodiimide hydrochloride (carbodiimide) | Millipore-Sigma | Catalog No. 03449 | |
| Chemical compound, drug | Glutaraldehyde 25% solution | Electron Microscopy Sciences | Catalog No. 16220 | |
| Chemical compound, drug | Osmium tetroxide 4% solution | Electron Microscopy Sciences | Catalog No. 18459 | |
| Chemical compound, drug | Paraformaldehyde 16% solution | Electron Microscopy Sciences | Catalog No. 15170 | |
| Chemical compound, drug | EMbed 812 | Electron Microscopy Sciences | Catalog No. 14120 | |
| Chemical compound, drug | Sodium cacodylate trihydrate | Electron Microscopy Sciences | Catalog No. 12300 | |
| Chemical compound, drug | Uranyl acetate | SPI-Chem | Catalog No. 02624AB | |
| Chemical compound, drug | DAPI | ThermoFisher Scientific | Catalog No. 1306 | IHC (3 µg/ml) |
| Chemical compound, drug | Mouse VEGF | Millipore-Sigma | Catalog No. V4512 | |
| Software, algorithm | LIMMA | *Ritchie et al., 2015* | RRID: SCR_010943 | |
| Other | D-GLUCOSE (U-13C6, 99%) | Cambridge Isotope Laboratories | Catalog No. CLM-1396-PK | |
| Other | Mixed fatty acids (U-D, 96–98%) | Cambridge Isotope Laboratories | Catalog No. DLM-8572-PK | |
| Other | Mixed fatty acids (13C, 98%+) | Cambridge Isotope Laboratories | Catalog No. CLM-8455-PK | |

## Immunohistochemical studies on human glioma specimens

Frozen surgical glioma specimens were obtained from the UCLA Department of Neurosurgery. Frozen autopsy control brain samples (frontal lobe, occipital lobe, and cerebellum) were obtained from the UCLA Section of Neuropathology. Samples were sectioned to 8 µm and placed on glass slides. All samples were fixed with 3% paraformaldehyde (PFA) in PBS/Ca/Mg and permeabilized in 0.2% Triton X-100 in PBS/Ca/Mg. Tissues were blocked with PBS/Ca/Mg containing 5% donkey serum and 0.2% bovine serum albumin (BSA) and incubated overnight at 4°C with one or more mouse monoclonal antibodies (mAbs) against human GPIHBP1 (RF4, RE3, RG3; 10 µg/ml) (*Hu et al., 2017*), a rabbit polyclonal antibody against von Willebrand factor (vWF) (Dako; 1:200), and a goat polyclonal antibody against human glial fibrillary acidic protein (GFAP) (Abcam; 1:500). In some experiments, recombinant soluble human GPIHBP1 (50 µg) was added to the primary antibody incubation. After washing the slides, 1-hr incubations were performed with an Alexa Fluor 647–conjugated

donkey anti–mouse IgG (ThermoFisher Scientific; 1:500), an Alexa Fluor 488–conjugated donkey anti–rabbit IgG (ThermoFisher Scientific; 1:500), and an Alexa Fluor 568–conjugated donkey anti–goat IgG (ThermoFisher Scientific; 1:500). DNA was stained with 4′,6-diamidino-2-phenylindole (DAPI). Images were taken with an LSM700 confocal microscope with an Axiovert 200M stand and processed with Zen 2010 software (Zeiss).

Immunoperoxidase staining was performed with the ImmPRESS Excel Staining Kit (Vector Laboratories). Endogenous peroxidase activity was quenched with BLOXALL Blocking Solution (Vector Laboratories). After incubating sections in 10% normal horse serum, sections were incubated for 1 hr with mAb RF4 (5 µg/ml), followed by a 15-min incubation with a goat anti-mouse IgG (10 µg/ml, Vector Laboratories). Slides were then incubated for 30 min with a horseradish peroxidase–conjugated horse anti–goat IgG (ImmPRESS Excel Reagent, Vector Laboratories). After washing, the slides were incubated with ImmPACT DAB EqV (Vector Laboratories) until a color change was evident (~30 s). Finally, sections were counterstained with hematoxylin and mounted with Vectashield Mounting Media (Vector Laboratories). Images were recorded with a Nikon Eclipse E600 microscope (Plan Fluor 40×/0.50 NA or 100×/0.75 NA objectives) equipped with a DS-Fi2 camera (Nikon).

## Genome dataset and gene-expression analyses

Cohorts for RNA-seq analysis were obtained from two databases: The Cancer Genome Atlas (TCGA) for tumor samples and Genotype-Tissue Expression (GTEx) for normal brain samples. Samples from TCGA (n = 157) and GTEx (n = 283) were processed with the TOIL pipeline as described (*Vivian et al., 2017*). A differential expression analysis of fatty acid metabolism genes was carried out with a linear model RNA-seq analysis software (LIMMA) (*Ritchie et al., 2015*). Genes were considered differentially expressed if the *p*-values were <0.05 and the $\log_2$ changes were >twofold. A heatmap was generated with the software R (*Kolde, 2015*).

## Animal procedures and glioma implantation

Mice on a C57BL/6 background expressing both the ROSA$^{mT/mG}$ *Cre*-reporter (*Muzumdar et al., 2007*) and tamoxifen-inducible Pdgfb-iCreER$^{T2}$ alleles (*Claxton et al., 2008*) were generated by breeding. In those mice, the administration of tamoxifen induces *Cre* recombinase expression in *Pdgfb*-positive cells. The recombination event results in the expression of EGFP in endothelial cells; all other cells express tdTomato. For the glioma implantation studies, mice (8–12-weeks-old) were injected intraperitoneally with tamoxifen (65 µg/g body weight, 4 injections in 2 weeks) before surgery. Mice were anesthetized with ketamine/xylazine, and a craniotomy was performed by drilling a 5-mm hole between the lambdoid, sagittal, and coronal sutures. A blue fluorescent protein (BFP)-tagged CT-2A glioblastoma spheroid (250-µm in diameter) (*Seyfried et al., 1992*; *Oh et al., 2014*) was injected into the cortex and sealed by cementing a glass coverslip on the skull. The CT-2A cell line was generated by Seyfried and coworkers through chemical induction with 20-methylcholanthrene in the brain of C57BL/6 mice and has been characterized extensively (*Seyfried et al., 1992*). In other experiments, CT-2A glioblastoma spheroids were implanted into the cortex of C57BL/6 wild-type mice and *Gpihbp1*$^{-/-}$ mice (*Beigneux et al., 2007*). Those procedures were performed as described previously (*Stanchi et al., 2019*).

## Immunohistochemical studies on mouse gliomas

Mice harboring BFP-expressing CT-2A gliomas (*Seyfried et al., 1992*; *Oh et al., 2014*) were anesthetized with ketamine/xylazine and then injected intravenously (*via* the tail vein) with 100 µg of an Alexa Fluor 647–conjugated antibody against mouse GPIHBP1 (11A12) (*Beigneux et al., 2009*). After 1 min, the mice were perfused through the heart with 15 ml of PBS, followed by 10 ml of 2% PFA in PBS. Brain and glioma tissues were harvested and fixed overnight in 4% PFA. Tissue sections (200-µm-thick) were prepared with a vibratome. For immunofluorescence microscopy studies, the sections were fixed with 4% PFA in PBS and blocked and permeabilized in TNBT (0.1 M Tris, pH 7.4, 150 mM NaCl, 0.5% blocking reagent from Perkin Elmer, 0.5% Triton X-100) for 4 hr at room temperature. Tissues were incubated with an antibody against GLUT1 (Millipore; 1:200) diluted in TNBT buffer overnight at 4°C, washed in TNT buffer (0.1 M Tris pH 7.4; 150 mM NaCl, 0.5% Triton X-100) and incubated with an Alexa Fluor 488–conjugated donkey anti–rabbit IgG (ThermoFisher Scientific;

1:200). Tissues were washed and mounted in fluorescent mounting medium (Dako). Images were obtained with a Leica TCS SP8 confocal microscope.

For the analysis of tissues from mice that were not injected with anti-GPIHBP1 antibodies, tissues were embedded in OCT medium, and 10-μm sections were cut with a cryostat. Sections were fixed with 3% PFA in PBS/Ca/Mg, permeabilized with 0.2% Triton X-100 in PBS/Ca/Mg, and blocked with PBS/Ca/Mg containing 5% donkey serum and 0.2% BSA. Tissue sections were incubated overnight at 4°C with a rabbit antibody against CD31 (Abcam; 1:50), a goat antibody against mouse LPL (12 μg/ml) (*Page et al., 2006*), an Alexa Fluor 488–conjugated antibody against F4/80, or an Alexa Fluor 647–conjugated antibody against mouse GPIHBP1 (11A12, 10 μg/ml). After removing non-bound antibodies and washing the sections, unlabeled primary antibodies were detected with an Alexa Fluor 488–conjugated donkey anti–rabbit IgG (ThermoFisher Scientific; 1:500) or an Alexa Fluor 568–conjugated donkey ant–goat IgG (ThermoFisher Scientific; 1:500). DNA was stained with DAPI, and tissues were mounted with ProLong Gold mounting media (ThermoFisher Scientific). Images were recorded on an LSM 800 confocal microscope (Zeiss).

## Immunocytochemistry studies on mouse peritoneal macrophages

Macrophages were collected by peritoneal lavage of C57BL/6 wild-type and $Lpl^{-/-}$MCK-hLPL mice. Cells were centrifuged at 400 × g for 5 min at 4°C, washed with 5 ml of red blood cell lysing buffer (Sigma) for 5 min, washed twice with cold PBS, and then plated onto FBS-coated Petri dishes. Cells were cultured overnight in macrophage medium (Dulbecco Modified Eagle Medium with 10% FBS, 1% glutamine, and 1% sodium pyruvate). On the next day, macrophages were lifted with cold PBS containing 5 mM EDTA for 30 min at 4°C. Cells were then plated onto poly-D-lysine–coated glass coverslips (75,000 cells/coverslip) and incubated overnight in macrophage media. On the following day, the cells were washed three times for 10 min in PBS/Ca/Mg containing 0.2% BSA and then incubated with Alexa Fluor 568–labeled ALO-D4 (a modified cytolysin that binds to 'accessible cholesterol' in the plasma membrane) (*Das et al., 2014*; *Das et al., 2013*; *Gay et al., 2015*) for 2 hr at 4°C. Samples were washed three times for 1 min with PBS/Ca/Mg, fixed with 3% PFA in PBS/Ca/Mg, permeabilized with 0.2% Triton X-100 in PBS/Ca/Mg, and blocked with PBS/Ca/Mg containing 5% donkey serum and 0.2% BSA. Cells were then incubated with a goat antibody against mouse LPL (12 μg/ml) (*Page et al., 2006*) for 1 hr at room temperature followed by a 30-min incubation with an Alexa Fluor 647–labeled donkey anti–goat IgG (ThermoFisher Scientific; 1:500). DNA was stained with DAPI, and coverslips were mounted onto glass slides in ProLong Gold mounting media (ThermoFisher Scientific). Images were recorded with a Zeiss LSM700 confocal microscope.

## Administration of [$^{13}$C]fatty acids, [$^{13}$C]glucose, and [$^{2}$H]TRLs to mice

C57BL/6 mice with CT-2A gliomas (of three-week duration) were given 80 μl of [$^{13}$C]fatty acids (~1 mg/μl; Cambridge Isotope Laboratories) or 80 μl of [$^{13}$C]glucose (3 mg/kg body weight; Cambridge Isotope Laboratories) by oral gavage every 12 hr for 36 hr (three doses). To study TRL metabolism, mice were injected intravenously with a single bolus of [$^{2}$H]TRLs (40 μg triglycerides in 100 μl) via the tail vein. The [$^{2}$H]TRLs were isolated from the plasma of $Gpihbp1^{-/-}$ mice after administering deuterated fatty acids by gastric gavage (*He et al., 2018a*). After allowing the [$^{2}$H]TRLs to circulate for 1 min or 30 min, the mice were perfused through the heart with 15 ml of ice-cold PBS/Ca/Mg at 3 ml/min (10 ml though the left ventricle and 5 ml through the right ventricle). Next, the mice were perfusion-fixed through the left ventricle with 10 ml of ice-cold 4% N-(3-dimethylaminopropyl)-N′-ethylcarbodiimide hydrochloride ('carbodiimide;' Sigma-Aldrich) (mass/vol) and 0.4% glutaraldehyde (Electron Microscopy Sciences) (vol/vol) in 0.1 M phosphate buffer. The heart, brain, and glioma tumors were collected and placed in 0.1 M phosphate buffer containing 4% carbodiimide and 2.5% glutaraldehyde for 2 hr at 4°C. Tissues were cut into 1-mm$^3$ pieces and fixed overnight in 2.5% glutaraldehyde, 3.7% PFA, and 2.1% sucrose in 0.1 M sodium cacodylate (pH 7.4).

## Preparation of tissue sections for NanoSIMS imaging and electron microscopy

After fixation, 1-mm$^3$ pieces of tissue were rinsed three times (10 min each) in 0.1 M sodium cacodylate buffer (pH 7.4) and fixed with 2% OsO$_4$ (Electron Microscopy Sciences) in 0.1 M sodium cacodylate on ice for 90 min. The samples were rinsed three times (10 min each) with distilled water and

stained overnight with 2% uranyl acetate at 4°C. On the following day, the samples were rinsed three times for 10 min each with distilled water and then dehydrated with increasing amounts of ethanol (30%, 50%, 70%, 85%, 95%, and 100%; $3 \times 10$ min) before infiltration with Embed812 resin (Electron Microscopy Sciences) diluted in acetone (33% for 2 hr; 66% overnight; 100% for 3 hr). The samples were embedded in polyethylene molds (Electron Microscopy Sciences) with fresh resin and polymerized in a vacuum oven at 65°C for 48 hr. The polymerized blocks were then removed from the molds, trimmed, and sectioned.

For transmission electron microscopy, 65-nm sections were cut and collected on freshly glow-discharged copper grids (Ted Pella) that were coated with formvar and carbon. Sections were then stained with Reynold's lead citrate solution for 10 min. Images were acquired with an FEI T12 transmission electron microscope set to 120 kV accelerating voltage and a Gatan 2K × 2K digital camera (Electron Imaging Center).

For NanoSIMS analyses, 500-nm sections were cut with a Leica UC6 ultramicrotome and collected on silicon wafers. Sections of tissue were coated with ~5 nm of platinum and analyzed with Nano-SIMS 50L or NanoSIMS 50 instruments (CAMECA). Samples were scanned with a 16-KeV $^{133}$Cs$^+$-beam, and secondary electrons (SEs) and secondary ions ($^1$H$^-$, $^2$H$^-$, $^{12}$C$^-$, $^{13}$C$^-$, $^{12}$C$^{14}$N$^-$) were collected. A $50 \times 50$-μm region of the section was pre-sputtered with a ~1.2-nA beam current (primary aperture D1 = 1) to reach a dose of ~$1 \times 10^{17}$ ions/cm$^2$ to remove the platinum coating and implant $^{133}$Cs$^+$ in order to ensure a steady state of secondary ion release. A ~$40 \times 40$-μm region was imaged with an ~3-pA beam current (primary aperture D1 = 2) and a dwell time of ~10 ms/pixel per frame for multiple frames. Both $256 \times 256$–pixel and $512 \times 512$–pixel images were obtained. Images were prepared using the OpenMIMS plugin in ImageJ. For image quantification, $^2$H/$^1$H and $^{13}$C/$^{12}$C ratios in the regions of interests were calculated with the OpenMIMS plugin and processed by GraphPad Prism 7.0.

## Tumor studies in wild-type and *Gpihbp1*-deficient mice

Three-month-old C57BL/6 wild-type (five females, six males) and *Gpihbp1*$^{-/-}$ mice (six females, five males) were injected intracranially with CT-2A glioma cells stably expressing a *Gaussia* luciferase reporter gene ($4 \times 10^5$ cells/mouse). Cells were injected 1 mm posterior and 2 mm lateral to the bregma at a depth of 2 mm. Tumor burden was monitored every three days by measuring *Gaussia* luciferase in the blood (*Mai et al., 2017*; *Tannous, 2009*). Mice were weighed at weekly intervals and were euthanized when they lost >20% of their body weight. After the mice were euthanized, their tumors and brains were weighed. All studies were approved by UCLA's Animal Research Committee.

## *Gaussia* luciferase measurements

To measure the levels of secreted *Gaussia* luciferase (sGluc), blood was obtained from the tail vein of mice and mixed with 50 mM EDTA to prevent coagulation. 5 μl of blood was transferred to a 96-well plate, and sGluc activity was measured by chemiluminescence after injecting 100 μl of 100 μM coelentarazine (Nanolight) (*Mai et al., 2017*; *Tannous, 2009*). Data were plotted as relative light units (RLU).

## Quantifying mouse and human transcripts by qRT-PCR

C57BL/6 wild-type mice and *Lpl*$^{-/-}$MCK-hLPL mice were anesthetized with isoflurane and perfused with PBS. Heart, brown adipose tissue (BAT), and quadricep were harvested and flash-frozen in liquid nitrogen. RNA was isolated with TRI reagent (Molecular Research), and quantitative (q)RT-PCR measurements were performed in triplicate with a 7900HT Fast real-time PCR system (Applied Biosystems). Gene expression was calculated with a comparative CT method and normalized to levels of cyclophilin A expression. Primers for mouse *Gpihbp1*, mouse *Lpl*, and human *LPL* are described in the 'Key Resources Table'.

## VEGF treatment of brain endothelial cells

Mouse brain microvascular endothelial cells (bEnd.3; ATCC #CRL-2299) were plated into 6-well plates and grown in DMEM media containing 10% FBS, 1% glutamine, and 1% sodium pyruvate overnight. On the next day, cells were rinsed with PBS and incubated in medium containing

recombinant mouse VEGF (100 ng/ml; Sigma) for another 24 hr. RNA was isolated with TRI reagent (Molecular Research), and qRT-PCR measurements were performed in triplicate with a 7900HT Fast real-time PCR system (Applied Biosystems). Gene expression was calculated with a comparative CT method and normalized to cyclophilin A expression. Primers for mouse *Gpihbp1*, *Cd31*, *Angpt2*, *Cxcr4*, and *Dusp5* are described in the 'Key Resources Table'.

## Cell lines

CT-2A cells were obtained originally from the Seyfried laboratory and have been extensively tested and characterized (*Seyfried et al., 1992*). These cells also robustly expressed GFAP. The bEnd.3 cells were obtained from ATCC with a proper 'certificate of analysis'. All cell lines were negative for mycoplasma contamination.

## Statistics

Statistical analyses of data were performed with GraphPad Prism 7.0 software. All data are shown as the means ± standard deviations. Differences were assessed using a Student's *t*-test with Welch's correction.

## Study approval

All tissue samples from patients were obtained after informed consent and with approval from the UCLA Institutional Review Board (IRB; protocol 10–000655). Animal housing and experimental protocols were approved by UCLA's Animal Research Committee (ARC; 2004-125-51, 2016–005) and the Institutional Animal Care and Research Advisory Committee of the KU Leuven (085/2016). The animals were housed in an AAALAC (Association for Assessment and Accreditation of Laboratory Animal Care International)-approved facility and cared for according to guidelines established by UCLA's Animal Research Committee. The mice were fed a chow diet and housed in a barrier facility with a 12 hr light-dark cycle.

# Acknowledgements

This work was supported by grants from the NHLBI (HL090553, HL087228, HL125335), a Transatlantic Network Grant from the Leducq Foundation (12CVD04), and the Belgian Cancer Foundation (Stichting Tegen Kanker, grant 2012-181, 2018–074). Xuchen Hu was supported by a Ruth L Kirschstein National Research Service Award (T32HL69766) and by the UCLA Medical Scientist Training Program. Ken Matsumoto was supported by a Japan Society for the Promotion of Science Postdoctoral Fellowship. Linda Liau was supported by a NCI Brain Tumor SPORE grant (P50-CA211015). Haibo Jiang was supported by an Australian Research Council Discovery Early Career Researcher Award and a Cancer Council Western Australia Early Career Investigator Grant. We thank Andre Bensadoun (Cornell University) for goat anti-mouse LPL antibodies; we also thank Anna Nowak (University of Western Australia) and the late Ben Barres (Stanford University) for helpful discussions.

# Additional information

## Competing interests

Holger Gerhardt: Reviewing editor, *eLife*. The other authors declare that no competing interests exist.

## Funding

| Funder | Grant reference number | Author |
| --- | --- | --- |
| National Heart, Lung, and Blood Institute | HL090553 | Stephen G Young |
| National Heart, Lung, and Blood Institute | HL087228 | Stephen G Young |
| National Heart, Lung, and Blood Institute | HL125335 | Stephen G Young |

| Fondation Leducq | 12CVD04 | Stephen G Young |
|---|---|---|
| National Institutes of Health | Ruth L Kirschstein National Research Service Award (T32HL69766) | Xuchen Hu |
| National Institute of General Medical Sciences | GM008042 | Xuchen Hu |
| National Cancer Institute | Brain Tumor SPORE grant (P50-CA211015) | Linda M Liau |
| Stichting Tegen Kanker | 2012-181 | Holger Gerhardt |
| Stichting Tegen Kanker | 2018-074 | Holger Gerhardt |
| Japan Society for the Promotion of Science | Postdoctoral Fellowship | Ken Matsumoto |
| University of California, Los Angeles | Medical Scientist Training Program | Xuchen Hu |
| Australian Research Council | Discovery Early Career Researcher Award | Haibo Jiang |
| Cancer Council Western Australia | Early Career Investigator Grant | Haibo Jiang |

The funders had no role in study design, data collection and interpretation, or the decision to submit the work for publication.

## Author contributions

Xuchen Hu, Conceptualization, Formal analysis, Validation, Investigation, Visualization, Writing—original draft, Writing—review and editing; Ken Matsumoto, Investigation, Visualization, Writing—review and editing; Rachel S Jung, Patrick J Heizer, Cuiwen He, Norma P Sandoval, Yiping Tu, Rochelle M Ellison, Jazmin E Morales, Lynn J Baufeld, Investigation; Thomas A Weston, Christopher M Allan, Investigation, Writing—review and editing; Harry V Vinters, Linda M Liau, Resources; Nicholas A Bayley, Data curation, Formal analysis; Liqun He, Formal analysis; Christer Betsholtz, Anne P Beigneux, Supervision, Writing—review and editing; David A Nathanson, Resources, Supervision, Writing—review and editing; Holger Gerhardt, Resources, Supervision, Funding acquisition, Writing—review and editing; Stephen G Young, Conceptualization, Resources, Formal analysis, Supervision, Funding acquisition, Methodology, Writing—original draft, Project administration, Writing—review and editing; Loren G Fong, Conceptualization, Resources, Formal analysis, Supervision, Methodology, Writing—original draft, Project administration, Writing—review and editing; Haibo Jiang, Resources, Formal analysis, Investigation, Writing—review and editing

## Author ORCIDs

Xuchen Hu  https://orcid.org/0000-0002-0944-624X
Holger Gerhardt  http://orcid.org/0000-0002-3030-0384
Stephen G Young  https://orcid.org/0000-0001-7270-3176
Loren G Fong  https://orcid.org/0000-0002-4465-5290
Haibo Jiang  https://orcid.org/0000-0002-2384-4826

## Ethics

Human subjects: All tissue samples from patients were obtained after informed consent and with approval from the UCLA Institutional Review Board (IRB; protocol 10-000655).

Animal experimentation: Animal housing and experimental protocols were approved by UCLA's Animal Research Committee (ARC; 2004-125-51, 2016-005) and the Institutional Animal Care and Research Advisory Committee of the KU Leuven (085/2016). The animals were housed in an AAALAC (Association for Assessment and Accreditation of Laboratory Animal Care International)-approved facility and cared for according to guidelines established by UCLA's Animal Research Committee.

**Decision letter and Author response**
Decision letter https://doi.org/10.7554/eLife.47178.038
Author response https://doi.org/10.7554/eLife.47178.039

## Additional files

### Supplementary files
• Transparent reporting form
DOI: https://doi.org/10.7554/eLife.47178.034

### Data availability
All data generated during this study are included in the manuscript and supporting files.

The following previously published dataset was used:

| Author(s) | Year | Dataset title | Dataset URL | Database and Identifier |
|---|---|---|---|---|
| He L, Vanlande-wijck M, Mae MA, Andrae J, Ando K, Del Gaudio F | 2018 | Molecular atlas of vascular and vessel-associated cell types in the mouse brain and lung | https://www.ncbi.nlm.nih.gov/geo/query/acc.cgi?acc=GSE98816 | NCBI Gene Expression Omnibus, GSE98816 |

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
