## [Decision Letter]

Thank you for submitting your article "GPIHBP1 expression in gliomas promotes utilization of lipoprotein-derived nutrients" for consideration by *eLife*. Your article has been reviewed by three peer reviewers, and the evaluation has been overseen by a Reviewing Editor and Harry Dietz as the Senior Editor. The following individuals involved in review of your submission have agreed to reveal their identity: Rodney Elwood Infante (Reviewer #1); Rolf Brekken (Reviewer #3).

The reviewers have discussed the reviews with one another and the Reviewing Editor has drafted this decision to help you prepare a revised submission.

Summary:

This well-presented study proposes an intriguing role for GPIHBP1-mediated LPL activity in the context of brain tumors. The authors show that GPIHBP1 is expressed in capillary endothelial cells of gliomas, but not in normal brains. Using NanoSIMS technology, at which the members of this group are experts, they show that GPIHBP1's interaction with LPL allows for gliomas to have increased triglyceride-rich lipoprotein uptake and utilization. They postulate that gliomas use this system as a source of nutrients for their growth. All reviewers agreed that the study was well-executed and that the findings were potentially significant, but all of them raised the same concern about whether GPIHBP1 was important for glioma development or progression. They welcomed a revised submission which addressed this concern, along with answers to some additional questions.

Essential revisions:

1) Please compare the tumor growth, number, and size in WT and GPIHBP1-deficient mice. If gliomas grow well in the absence of GPIHBP1, is this due to expression of Glut-1 on the vasculature in gliomas in GPIHBP1-deficient mice?

2) Provide some additional insight into the mechanism of increased GPIHBP1 expression in glioma associated vasculature. What is known about GPIHBP1 expression in other tissues and are any factors known to be associated with its expression?

3) Is it possible that the tail-vein injected Alexa Fluor 647-labeled GPIHBP1 antibody has access to gliomas and not to the normal brain tissue because of an impaired blood brain barrier in the case of the gliomas? Although there are figures where GPIHBP1 was stained after fixation, it is always done on slides with only normal tissue. An experiment similar to what is done for Glut1 in Figure 4 should be done for GPIHBP1 – slides containing both glioma and normal tissue where both GLUT1 and GPIHBP1 antibodies are used after fixation.

4) Does the expression of LPL affect the expression of GPIHBP1 in gliomas? The LPL knockout mouse model was used for IHC for normal brain tissue, but not for the glioma model (Figure 5—figure supplement 5).

[Editors' note: further revisions were requested prior to acceptance, as described below.]

Thank you for resubmitting your work entitled "GPIHBP1 expression in gliomas promotes utilization of lipoprotein-derived nutrients" for further consideration at *eLife*. Your revised article has been favorably evaluated by Harry Dietz (Senior Editor) and a Reviewing Editor.

The manuscript has been improved but there are some remaining issues that need to be addressed before acceptance, as outlined below:

The authors have addressed most of the points raised by the reviewers by providing new supporting data and reframing parts of the manuscript. The reviewers all agreed that this revised manuscript will be suitable for publication once one lingering concern is addressed. The authors clearly show that GPIHBP1 is not critical to glioma progression, at least in implant models (Figure 8—figure supplement 4), and provide a satisfactory explanation for why this may be so. Consequently, statements implying that GPIHBP1 expression in gliomas is important for tumor growth should be toned down or removed (e.g. in the impact statement and the last sentence of the Abstract). The authors may instead choose to highlight how the stark difference in GPIHBP1 expression in normal versus glioma capillaries could be utilized as a therapeutic tool (as they describe in the last paragraph of the Discussion).

---

## [Author Response]

Summary:This well-presented study proposes an intriguing role for GPIHBP1-mediated LPL activity in the context of brain tumors. The authors show that GPIHBP1 is expressed in capillary endothelial cells of gliomas, but not in normal brains. Using NanoSIMS technology, at which the members of this group are experts, they show that GPIHBP1's interaction with LPL allows for gliomas to have increased triglyceride-rich lipoprotein uptake and utilization. They postulate that gliomas use this system as a source of nutrients for their growth. All reviewers agreed that the study was well-executed and that the findings were potentially significant, but all of them raised the same concern about whether GPIHBP1 was important for glioma development or progression. They welcomed a revised submission which addressed this concern, along with answers to some additional questions.

We thank the reviewers and editors for their kind and generous comments about our manuscript. Over the past month, we have performed additional experiments in order to respond to the criticisms of the reviewers. In the revised manuscript, we believe that we have addressed all of the reviewers’ concerns. In addition, we added an additional experiment in which we performed NanoSIMS imaging of heart and brain of wild-type and *Gpihbp1*-deficient mice 15 min following the injection of [^2^H]TRLs (Figure 7—figure supplement 2).

Essential revisions:1) Please compare the tumor growth, number, and size in WT and GPIHBP1-deficient mice. If gliomas grow well in the absence of GPIHBP1, is this due to expression of Glut-1 on the vasculature in gliomas in GPIHBP1-deficient mice?

We thank the reviewers for this suggestion. We injected CT-2A glioma cells that had been stably transfected with a Gaussia luciferase reporter (1, 2) into the brain of wild-type (n = 11) and *Gpihbp1*-deficient mice (n = 11). We monitored tumor mass by measuring the level of luciferase in the blood over a span of 3 weeks. In addition, we measured body weight at regular intervals, euthanizing mice when they lost more than 20% of their initial body weight. Finally, we weighed tumors after euthanizing the mice. We observed no statistically significant differences in tumor growth, tumor size, or survival between wild-type and *Gpihbp1*^–/–^ mice. In the revised manuscript, we have included a supplemental figure showing these data (Figure 8—figure supplement 4).

The fact that we did not find significantly smaller tumors in *Gpihbp1*^–/–^ mice was not surprising. Despite markedly elevated plasma triglyceride levels and inefficient processing of lipoproteins, *Gpihbp1*^–/–^ mice appear healthy and have normal cardiac function. The heart normally uses large amounts of lipoprotein-derived triglycerides for fuel but manages to function normally in *Gpihbp1*^–/–^ mice, likely because of the heart’s ability to compensate by utilizing glucose and/or albumin-bound fatty acids. Given those observations, we suspected that the gliomas in *Gpihbp1*^–/–^ mice would be able to grow fairly normally—as a result of their ability to use glucose and/or glucose-derived metabolites as fuel. When [^13^C]glucose was administered to wild-type mice, NanoSIMS imaging revealed substantial ^13^C enrichment in tumors. Also, our paper demonstrated that glioma capillary endothelial cells express GLUT1. In the revised manuscript, we show additional studies that document GLUT1 expression in glioma capillaries of wild-type and *Gpihbp1*^–/–^ mice (Figure 4—figure supplement 4).

Our studies provide strong support for the notion that GPIHBP1 expression in glioma capillaries facilitates uptake of lipoprotein-derived nutrients by tumor cells in wild-type mice. In the revised manuscript, we report that glioma tumors can grow in the absence of GPIHBP1, likely a result of the same sort of compensatory mechanisms that allow normal heart function in *Gpihbp1*-deficient mice.

2) Provide some additional insight into the mechanism of increased GPIHBP1 expression in glioma associated vasculature. What is known about GPIHBP1 expression in other tissues and are any factors known to be associated with its expression?

Thank you; this is an important issue. Thus far, mechanisms underlying GPIHBP1 expression in capillary endothelial cells are unclear. However, Chiu and coworkers recently reported that GPIHBP1 expression in cultured rat aortic endothelial cells can be induced by VEGF (3).

Over the past month, we have tested GPIHBP1 expression in the mouse brain endothelial cell line bEnd.3. We found that treating those cells with VEGF increased *Gpihbp1* transcripts substantially. We have included those studies as a supplementary figure in the revised manuscript (Figure 3—figure supplement 3).

The induction of *Gpihbp1* expression by VEGF could potentially explain why GPIHBP1 is expressed in glioma capillaries in vivo. Glioma cells are known to produce VEGF (4–6).

In terms of understanding *Gpihbp1* expression, we would also draw the reviewers’ attention to our recent discovery of a *Gpihbp1* enhancer element, located ∼3.6 kb upstream from exon 1 of mouse *Gpihbp1* (7). Mice with a deletion of this enhancer had a ~90% decrease in *Gpihbp1* expression in capillaries of the liver but only a ~50% decrease in *Gpihbp1* transcripts in capillaries of heart and brown adipose tissue. The impact of the enhancer deletion on *Gpihbp1* expression was insufficient to elicit hypertriglyceridemia. Our publication provided the first insight into DNA sequences regulating *Gpihbp1* expression.

3) Is it possible that the tail-vein injected Alexa Fluor 647-labeled GPIHBP1 antibody has access to gliomas and not to the normal brain tissue because of an impaired blood brain barrier in the case of the gliomas? Although there are figures where GPIHBP1 was stained after fixation, it is always done on slides with only normal tissue. An experiment similar to what is done for Glut1 in Figure 4 should be done for GPIHBP1 – slides containing both glioma and normal tissue where both GLUT1 and GPIHBP1 antibodies are used after fixation.

We agree that the blood–brain barrier of gliomas can be leaky. However, there is no reason to believe that intravenously injected Alexa Fluor 647–labeled GPIHBP1 antibody would have greater access to glioma vasculature, simply because GPIHBP1 is located on the luminal surface of capillary endothelial cells. Thus, the injected antibodies would not need to cross the blood–brain barrier to bind to GPIHBP1. To further address the reviewer’s concerns, we performed staining of glioma and normal brain tissue with antibodies against GPIHBP1 and GLUT1 *after tissue fixation* (Figure 4—figure supplement 2). GPIHBP1 was present in endothelial cells of tumors but not in normal brain; GLUT1 was present in endothelial cells in both tumor and adjacent normal brain.

4) Does the expression of LPL affect the expression of GPIHBP1 in gliomas? The LPL knockout mouse model was used for IHC for normal brain tissue, but not for the glioma model (Figure 5—figure supplement 5).

We have no evidence that the expression of LPL influences the expression of GPIHBP1. GPIHBP1 and LPL in peripheral tissues are produced in different cell types—GPIHBP1 in endothelial cells and LPL in parenchymal cells. We no longer have access to adult LPL knockout mice. However, to address the reviewers’ concern, we examined transcript levels in archived tissue biopsies (heart, brown adipose tissue, skeletal muscle) from wild-type and *Lpl^–/–^*MCK-hLPL mice. The MCK-hLPL transgene is expressed at high levels in skeletal muscle and at much lower levels in the heart; the transgene is not expressed in brown adipose tissue. In these studies, our goal was to determine whether *Gpihbp1* expression in skeletal muscle or heart would be altered by overexpression of human LPL. In heart and brown adipose tissue, *Gpihbp1* expression levels were similar in wild-type mice and the *Lpl^–/–^*MCK-hLPL mice. In skeletal muscle, where hLPL levels are very high, *Gpihbp1* expression was also similar in wild-type mice and the *Lpl^–/–^*MCK-hLPL mice. Thus, overexpression of LPL did not influence *Gpihbp1* expression (Figure 5—figure supplement 8).

We would add that we recently documented normal plasma GPIHBP1 levels (a measurement that reflects GPIHBP1 expression in capillaries) in a human subject who was homozygous for a nonsense mutation in *LPL* (8), consistent with the notion that the amount of LPL expression does not affect GPIHBP1 expression in capillary endothelial cells.

In light of these findings, there is little reason to believe that the modest amounts of LPL expression in the brain would influence GPIHBP1 expression in gliomas.

**References**

1. Mai, W. X., L. Gosa, V. W. Daniels, L. Ta, J. E. Tsang, B. Higgins, W. B. Gilmore, N. A. Bayley, M. D. Harati, J. T. Lee, W. H. Yong, H. I. Kornblum, S. J. Bensinger, P. S. Mischel, P. N. Rao, P. M. Clark, T. F. Cloughesy, A. Letai, and D. A. Nathanson. 2017. Cytoplasmic p53 couples oncogene-driven glucose metabolism to apoptosis and is a therapeutic target in glioblastoma. Nat Med 23: 1342–1351.

2. Tannous, B. A. 2009. Gaussia luciferase reporter assay for monitoring biological processes in culture and in vivo. Nat Protoc 4: 582–591.

3. Chiu, A. P., A. Wan, N. Lal, D. Zhang, F. Wang, I. Vlodavsky, B. Hussein, and B. Rodrigues. 2016. Cardiomyocyte VEGF regulates endothelial cell GPIHBP1 to relocate lipoprotein lipase to the coronary lumen during diabetes mellitus. Arterioscler Thromb Vasc Biol 36: 145–155.

4. Plate, K. H., G. Breier, H. A. Weich, H. D. Mennel, and W. Risau. 1994. Vascular endothelial growth factor and glioma angiogenesis: coordinate induction of VEGF receptors, distribution of VEGF protein and possible in vivo regulatory mechanisms. Int J Cancer 59: 520–529.

5. Pietsch, T., M. M. Valter, H. K. Wolf, A. von Deimling, H. J. Huang, W. K. Cavenee, and O. D. Wiestler. 1997. Expression and distribution of vascular endothelial growth factor protein in human brain tumors. Acta Neuropathol 93: 109–117.

6. Christov, C., H. Adle-Biassette, C. Le Guerinel, S. Natchev, and R. K. Gherardi. 1998. Immunohistochemical detection of vascular endothelial growth factor (VEGF) in the vasculature of oligodendrogliomas. Neuropathol Appl Neurobiol 24: 29–35.

7. Allan, C. M., P. J. Heizer, Y. Tu, N. P. Sandoval, R. S. Jung, J. E. Morales, E. Sajti, T. D. Troutman, T. L. Saunders, D. A. Cusanovich, A. P. Beigneux, C. E. Romanoski, L. G. Fong, and S. G. Young. 2019. An upstream enhancer regulates Gpihbp1 expression in a tissue-specific manner. J Lipid Res 60: 869–879.

8. Beigneux, A. P., K. Miyashita, M. Ploug, D. J. Blom, M. Ai, M. F. Linton, W. Khovidhunkit, R. Dufour, A. Garg, M. A. McMahon, C. R. Pullinger, N. P. Sandoval, X. Hu, C. M. Allan, M. Larsson, T. Machida, M. Murakami, K. Reue, P. Tontonoz, I. J. Goldberg, P. Moulin, S. Charrière, L. G. Fong, K. Nakajima, and S. G. Young. 2017. Autoantibodies against GPIHBP1 as a cause of hypertriglyceridemia. N Engl J Med 376: 1647–1658.

[Editors' note: further revisions were requested prior to acceptance, as described below.]

The manuscript has been improved but there are some remaining issues that need to be addressed before acceptance, as outlined below:The authors have addressed most of the points raised by the reviewers by providing new supporting data and reframing parts of the manuscript. The reviewers all agreed that this revised manuscript will be suitable for publication once one lingering concern is addressed. The authors clearly show that GPIHBP1 is not critical to glioma progression, at least in implant models (Figure 8—figure supplement 4), and provide a satisfactory explanation for why this may be so. Consequently, statements implying that GPIHBP1 expression in gliomas is important for tumor growth should be toned down or removed (e.g. in the impact statement and the last sentence of the Abstract). The authors may instead choose to highlight how the stark difference in GPIHBP1 expression in normal versus glioma capillaries could be utilized as a therapeutic tool (as they describe in the last paragraph of the Discussion).

In the current revision, we have removed all statements implying the possibility that GPIHBP1 could be relevant to the growth of gliomas, including in the “impact statement.” Instead, we simply state that GPIHBP1 expression facilitates TRL processing in gliomas, providing a source of lipid nutrients for glioma cells.